# MPicker: visualizing and picking membrane proteins for cryo-electron tomography

Xiaofeng Yan [1,2,3,4,5], Shudong Li [5,6], Weilin Huang[1,2,3,4,5,6,7], Hao Wang[1,2,3,4,5], Tianfang Zhao[1,2,3,4,5,6,7], Mingtao Huang [1,2,3,4,5,6,7], Niyun Zhou[5], Yuan Shen [6,7] & Xueming Li [1,2,3,4,5] ✉

Advancements in cryo-electron tomography (cryoET) allow the structure of macromolecules to be determined in situ, which is crucial for studying membrane protein structures and their interactions in the cellular environment. However, membranes are often highly curved and have a strong contrast in cryoET tomograms, which masks the signals from membrane proteins. These factors pose difficulties in observing and revealing the structures of membrane proteins in situ. Here, we report a membrane-flattening method and the corresponding software, MPicker, designed for the visualization, localization, and orientation determination of membrane proteins in cryoET tomograms. This method improves the visualization of proteins on and around membranes by generating a flattened tomogram that eliminates membrane curvature and reduces the spatial complexity of membrane protein analysis. In MPicker, we integrated approaches for automated particle picking and coarse alignment of membrane proteins for sub-tomogram averaging. MPicker was tested on tomograms of various cells to evaluate the method for visualizing, picking, and analyzing membrane proteins.

Cryo-electron tomography (cryoET) is a powerful method for directly studying three-dimensional (3D) structures of organisms in their native state. Membranes are abundant in cells and are key platforms for various biological processes via embedded or associated proteins. Therefore, understanding the in situ structure and spatial distribution of membrane proteins is important[1–3]. However, recognizing and analyzing membrane proteins in cryoET tomograms is often challenging owing to the crowded environment of cells, low signal-to-noise ratio (SNR), and the missing wedge effect. Tomograms are usually viewed as a series of two-dimensional (2D) slices parallel to the lamellar sample, referred to as $x–y$ slices, which are generally perpendicular to the direction of the incident electron beam. The membranes are usually randomly oriented in a tomogram and hence do not match the structured voxel arrangement of tomogram data; thus, direct observation of membrane proteins, such as through $x–y$ slice images, is difficult. Furthermore, since the contrast of membranes in tomograms is substantially stronger than that of the embedded membrane proteins, it often masks the visible signals of membrane proteins. These factors limit the in situ structural analysis of membrane proteins for not only spatial organization but also high-resolution structure determination by sub-tomogram averaging (STA).

Visualization is often a prerequisite in cryoET for identifying and locating proteins, including proteins embedded in or associated with membranes. Some software packages, such as IMOD[4], have integrated tools to display a planar slice in a tomogram in an arbitrary direction, facilitating the visualization of membrane proteins distributed on a plane[5]. However, the curved nature of cell membranes makes visualization under a simple plane difficult. Membranorama[2] is a software

[1]Key Laboratory for Protein Sciences of Ministry of Education, School of Life Sciences, Tsinghua University, Beijing, China. [2]State Key Laboratory of Membrane Biology, School of Life Sciences, Tsinghua University, Beijing, China. [3]Tsinghua-Peking Joint Center for Life Sciences, Beijing, China. [4]Beijing Frontier Research Center for Biological Structure, Beijing, China. [5]School of Life Sciences, Tsinghua University, Beijing, China. [6]Department of Electronic Engineering, Tsinghua University, Beijing, China. [7]Beijing National Research Center for Information Science and Technology, Tsinghua University, Beijing, China. ✉e-mail: lixueming@tsinghua.edu.cn

specialized for membrane protein visualization, that offers an alternative approach for visualizing membrane proteins by mapping the molecular landscapes of membranes on a 3D membrane surface extracted from membrane segmentation, allowing the manual selection and orientation determination of membrane proteins.

In addition to visualization methods that assist manual particle picking, other image-feature-analysis-based methods, including template matching and deep-learning-based target detection that serve as tools for automated particle picking have also been developed. Some intrinsic properties of membrane proteins, such as unique distribution patterns different from those of soluble proteins, can be used for recognition. PySeg[6] utilizes membrane segmentation and automatically localizes membrane proteins by tracking a density network near the membrane. MemBrain[7] is a deep-learning-based method that utilizes the geometry of segmented membranes to extract and pre-align candidate subvolumes, thereby reducing the influence of membrane signals.

In this study, we develop a membrane-flattening-based method using the software MPicker for visualizing and localizing membrane proteins in a tomogram. MPicker can flatten a selected membrane in a tomogram by transforming the curved membrane and its adjacent density into a reformatted tomogram, termed the flattened tomogram. The fundamental concept of flattening is to reduce the dimensionality of the curved membrane and associated objects from 3D to 2D. This transformation allows for the convenient visualization and analysis of their spatial arrangements and enables the adoption of many conventional tools previously developed for 2D analysis. As a result, automated particle picking, 2D classification, and 3D particle alignment can be integrated into the MPicker workflow with minimal adaptation. In summary, a variety of analyses are supported on the flattened tomogram, including visualization, particle picking, and preliminary estimation of particle orientation. The key functional features of MPicker were demonstrated using diverse cell samples.

## Results
### Membrane flattening

Membrane flattening is the core process of MPicker, which supports the visualization and localization of proteins on the membrane surface. The thylakoid membrane in the *Chlamydomonas* tomogram[2] (Fig. 1a) was used to illustrate this procedure. The raw tomogram intended for flattening may first undergo denoising and missing wedge correction to enhance visualization[8], as well as membrane segmentation to generate a membrane mask. Membrane flattening comprises three steps: extracting the surface of a selected membrane, flattening the membrane surface to a plane, and extending the flattened region around the surface to produce a flattened tomogram.

In the first step, MPicker needs a set of starting points (red curve and green surface in Fig. 1b left panel) on the membrane surface to generate a curved surface expression by thin-plate spline (TPS) interpolation or polynomial fitting. If a membrane mask (Fig. 1a right panel) is provided, generated by either MPicker's build-in subroutine (Supplementary Fig. 1) or a third-party approach, the user is required to label one or multiple seed points on the mask to specify a membrane piece of interest. Subsequently, MPicker automatically extracts the points near the seed points on the mask to serve as starting points (see "Methods"). In the case of missing a membrane mask, for example, failing to perform segmentation on some low-quality membranes, manual labeling of the starting points on a membrane surface is an alternative way in MPicker. After obtaining a curved surface expression using the starting points, MPicker calculates and stores the normal vectors of the surface at each point for the third step and future orientation estimation.

In the second step, MPicker projects the extracted membrane surface on a best-fit cylindrical surface (cyan surface in Fig. 1b right panel) and further unrolls the cylindrical surface to a plane (yellow plane in Fig. 1b right panel). To accommodate various membrane shapes, MPicker supports two types of cylindrical surfaces: the open

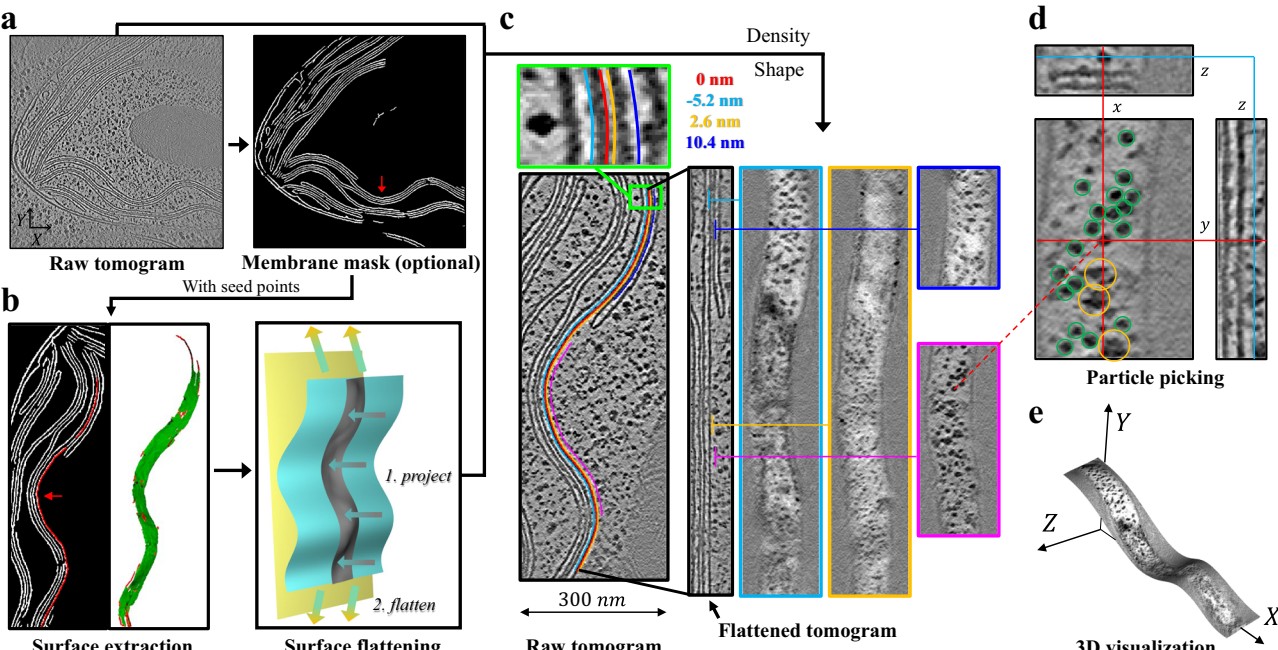

**Fig. 1 | Overview of MPicker. a** Input tomogram (EMD-10780) and membrane mask. A seed point (red arrow) is added to specify the membrane to be flattened. **b** Workflow to extract (left panel) and flatten (right panel) the surface. Several key elements are the starting points (red and green) on a membrane mask (white), the extracted membrane surface (gray), a cylindrical surface (cyan), and the final plane (yellow) for projection. **c** Membranes before and after flattening. Different colors are used to distinguish the position of the flattened surface in the raw tomogram, and the position of the x–y slices in the flattened tomogram. **d** Representative x–y, x–z, and y–z slices of the flattened tomogram. Picked particles of ATP synthase (green) and ribosome (orange) are labeled by circles. **e** 3D view of the cyan slice in (**c**). X–Y–Z indicates the coordinate system of the raw tomogram.

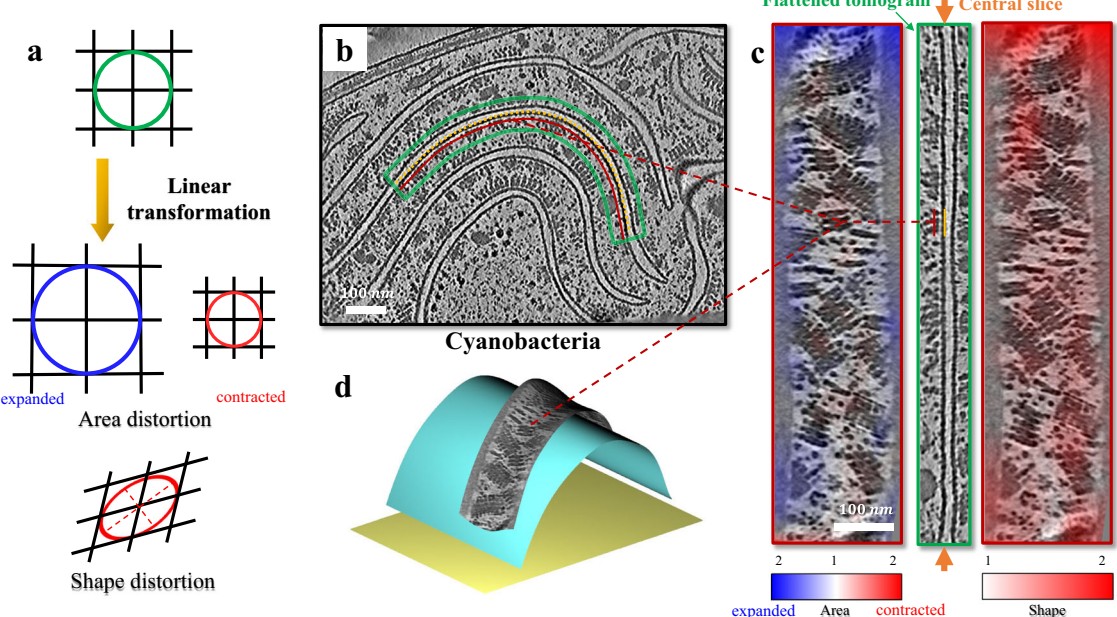

**Fig. 2 | Distortion analysis. a** Distortions induced by linear transformations. **b** A representative membrane region (green) to be flattened in a *Cyanobacterium* thylakoid tomogram (EMD-13771). The surfaces corresponding to the central slice and a parallel slice in (**c**) are indicated by an orange dotted curve and a red curve, respectively. **c** Distortions presented by color maps. The membrane region and the slice in a flattened tomogram are labeled with the same color as the corresponding elements in (**b**). **d** 3D view of the slice indicated by the red in both (**b**) and (**c**). The cylindrical surface (cyan) and the final plane (yellow) for projection are shown. The corresponding locations in different views are connected by red dashed lines.

type described by a polynomial, and the closed type described by an elliptic formula (see "Methods"). For example, the cyan surface in Fig. 1b is an open cylindrical surface described by an eighth-order polynomial. At the end of this step, pixel-level coordinate mapping between the membrane surface and the plane can be established. By plotting the membrane surface density on the plane, a flattened membrane surface image, referred to as the central slice, is generated.

In the third step, MPicker extends the central slice along both directions of the local normal vectors, generating a series of parallel slices spaced equally with a single pixel interval (Fig. 1c). All these slices constitute a new tomogram, termed the flattened tomogram. The *z*-axis of the flattened tomogram is perpendicular to the membrane, and each *x*–*y* slice corresponds to an isometric surface parallel to the membrane surface. In the flattened tomogram, all local cellular structures are preserved relative to the adjacent membrane. For example, the membrane retains its thickness, and proteins locally retain their orientation relative to the membrane. At the end of this step, the voxel-level coordinate mapping between the flattened and raw tomograms is calculated and stored.

The spatial organization of proteins relative to the original membrane is simplified with the flattened tomogram, supporting further analysis of protein distributions on the membrane, analysis of protein-membrane interactions, and particle picking (Fig. 1d). MPicker can also render a flattened tomogram slice as a curved surface, and thus depict the spatial relationships of objects in the original 3D space (Fig. 1e).

### Distortion caused by flattening

Membrane flattening involves a transformation from a curved surface to a plane, or say, mapping of an arbitrary surface to a plane, which is also termed surface parameterization in other fields[9]. Flattening can cause geometric distortions. For example, the geodesic distance between two points or the shape of an object on a curved surface may change after being mapped to a plane. The transformation from a cylindrical surface to a plane is a distortion-free intermediate flattening process. Distortion may occur only during the transformation from the

original curved surface to a cylindrical surface. In practice, few surfaces can be transformed into planes without distortion. Therefore, MPicker provides a quantitative evaluation of distortion to avoid the possible misinterpretation of structural features.

Generally, the distortions from surface parameterization (or flattening in this work) can be described by area and shape distortions. Intuitively, a unit circle may become an ellipse with a semi-major axis $\sigma_1$ and a semi-minor axis $\sigma_2$ after flattening (Fig. 2a). MPicker models flattening-induced distortion in a local region using a linear transformation (see "Methods"). The area distortion is quantified by the change in the area of the ellipse, $\sigma_1 \sigma_2$ (if larger than 1, records as expansion, else, takes the reciprocal and records as contraction). The shape distortion is quantified by the ratio between two axes, $\frac{\sigma_1}{\sigma_2}$.

We used a cyanobacterial thylakoid membrane piece[10] (green box in Fig. 2b) to demonstrate the flattening-induced distortion. The selected membrane surface (dotted orange line in Fig. 2b) was flattened and extended to generate a flattened tomogram (Fig. 2c). The selected membrane surface corresponded to the central slice in the flattened tomogram (orange arrows and a short line in Fig. 2c). We chose a slice parallel to the central slice (red short line in Fig. 2c) and calculated its distortion. The area distortion predominantly manifests as an expansion (blue color, Fig. 2c left) because the corresponding surface in the raw tomogram (red line in Fig. 2b) is located on the inner side of the curved central membrane surface (dotted orange line in Fig. 2b), and hence slightly stretched during flattening. MPicker also provides a visualization tool that maps a slice in the flattened tomogram back to the 3D space (Fig. 2d, Supplementary Fig. 2), which serves as a distortion-free reference for comparison.

Here, most regions on the selected slice show less than two-fold distortions for both area (Fig. 2c, left) and shape (Fig. 2c, right). Projecting the membrane surface on a cylindrical surface (Fig. 2d) is essential for minimizing distortion. Direct projection on a plane without cylindrical intermedium would cause significant distortion on both ends of the flattened membrane (Supplementary Fig. 2). However, for a complicated surface that deviates significantly from a cylindrical shape, using cylindrical intermedium might be insufficient for

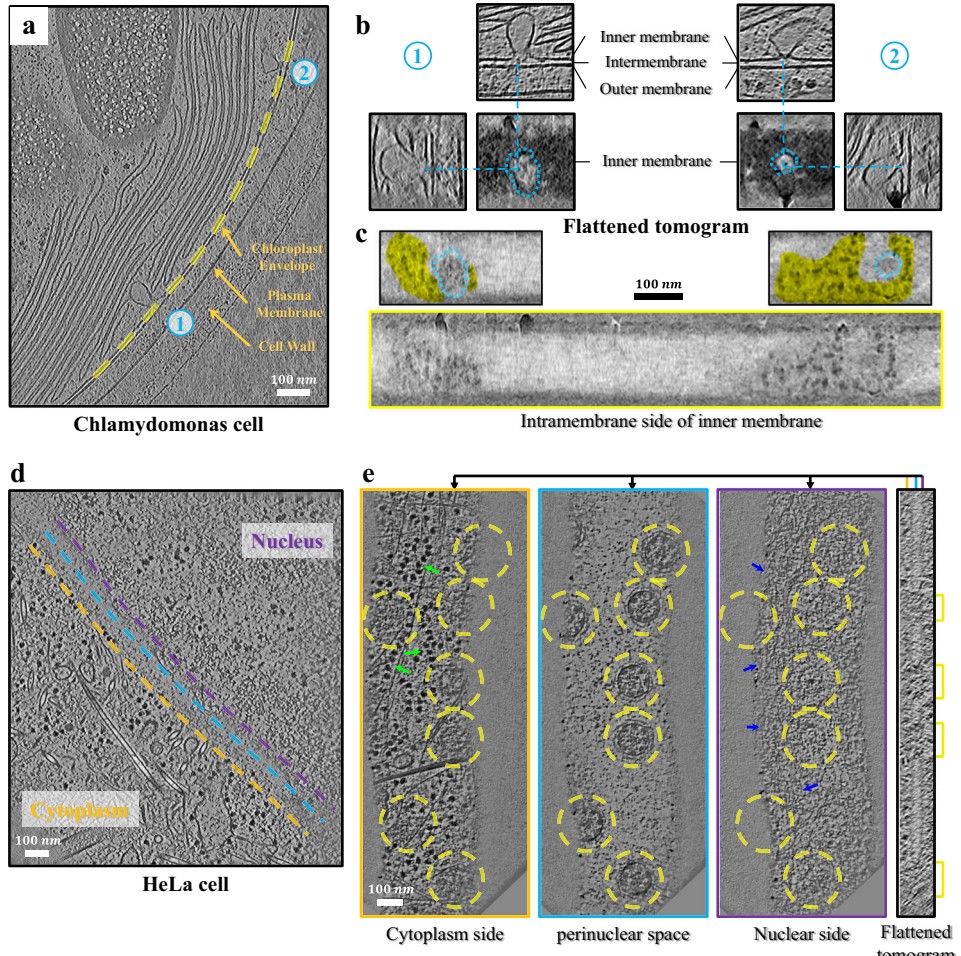

**Fig. 3 | Examples of the flattened tomograms. a** Representative view of a *Chlamydomonas* chloroplast tomogram. Two budding vesicles are labeled 1 and 2 (cyan). **b** $x$–$y$, $x$–$z$, and $y$–$z$ section views of the budding vesicles in a flattened tomogram. **c** $x$–$y$ slice in the flattened tomogram, corresponding to the intermembrane side of the inner membrane (yellow curve in **a**). In the subgraphs of two localized regions, the holes corresponding to budding regions are indicated by cyan circles, and the proteins near the holes are labeled in yellow. (**b**) and (**c**) share the same scale bar. **d** Representative view of a HeLa cell nuclear membrane tomogram (EMD-11992). **e** $x$–$y$ slices and a side section view of a flattened tomogram. Positions of $x$–$y$ slices are marked by lines with the same color as in (**d**) and (**e**). Nuclear pore complexes (NPCs; yellow circles), polyribosomes (green arrows), and putative lamina filaments (blue arrows) are pointed out in these $x$–$y$ slices. To enhance the contrast of the membrane and NPC (yellow), 61 $y$–$z$ slices (about 103 nm) of the flattened tomogram were summed to get the side section view.

flattening. An elegant approach is to use a triangle mesh as an intermedium, the parameterization of which has been extensively studied[9,11]. However, the generation of a triangle mesh usually requires a membrane mask and is highly dependent on membrane segmentation quality. Noises and segmentation errors affect the degree of matching between the triangle mesh and the original membrane and thus may cause failure in further flattening. Nevertheless, MPicker supports membrane flattening based on a given triangle mesh (see "Methods"). In a test using the *Chlamydomonas* endoplasmic reticulum (ER)[1], we generated a triangle mesh for a membrane with a complex shape and successfully flattened the membrane (Supplementary Fig. 3). The flattened tomogram reveals a clear distribution of ribosomes on the membrane surface.

### Membrane flattening on various membranes

To validate membrane flattening, MPicker was tested on the membranes of various cell samples, including *Chlamydomonas* chloroplasts, cyanobacteria, and HeLa cell nuclei.

Several thylakoid membranes were observed initially. In a tomogram of *Chlamydomonas* chloroplasts (EMD-10780, Fig. 1a), MPicker exhibits various proteins, such as photosystem II (PSII) complexes randomly distributed on the thylakoid membrane (cyan and blue slices

in Fig. 1c). In another tomogram of cyanobacteria (EMD-13771, Fig. 2b), several phycobilisome (PBS) complex strings attached to the thylakoid membrane are visualized in a flattened tomogram (Fig. 2c).

In another *Chlamydomonas* chloroplast tomogram, the inner chloroplast membrane with two budding vesicles (labeled as 1 and 2 in Fig. 3a) was flattened. Budding vesicles were ignored by excluding their starting points, so two holes appear at the budding sites in the flattened tomogram (Fig. 3b). Intriguingly, many protein particles were observed surrounding the budding sites on the intermembrane side of the inner membrane (Fig. 3c).

In the tomogram of HeLa cells[12] (Fig. 3d), the nuclear membrane nearly perpendicular to the $z$-axis was invisible owing to the missing wedge effect and could only be recognized indirectly by the fuzzy gap between the cytoplasm and the nucleoplasm. Generating a membrane mask is often difficult in such cases. Alternatively, we manually labeled 22 starting points along the center of the gap (cyan dashed line) and carried out flattening. In the flattened tomogram (Fig. 3e, Supplementary Movie 1), the nuclear membrane exhibited a weak signal and was flat. Nuclear pore complexes (NPCs, yellow circles) were observed in the membrane, and many polyribosomes on the cytoplasmic side (green arrows) and filaments on the nuclear side (blue arrows) were visible in slices of the flattened tomogram.

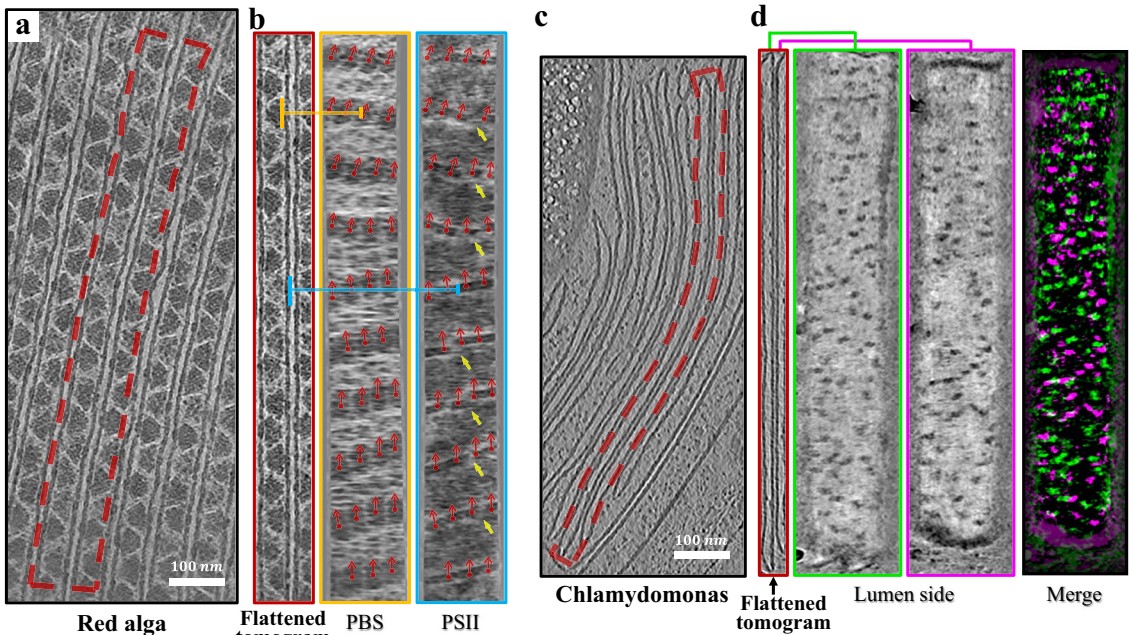

**Fig. 4 | Spatial relationship between particles on different flattened tomogram slices. a** Representative view of a red alga chloroplast tomogram (EMD-31243). **b** Flattened tomogram of the thylakoid membrane (red region in **a**). Two *x*–*y* slices (orange, cyan) on different sides of a membrane correspond to the vertical lines in the flattened tomogram. The positions and orientations of phycobilisome (PBS) particles are labeled by red arrows and copied to the slice showing PSII. The breaking points in PSII strings are marked by yellow arrows. (**a**) and (**b**) share the same scale bar. **c** Representative view of *Chlamydomonas* tomogram data as shown in Fig. 3a. **d** Flattened tomogram of a thylakoid membrane pair (red region in **c**). Two *x*–*y* slices (green, and magenta) show the lumen side density of two nearby membranes, respectively. Particles from the two slices were drawn with different colors and shown on a merged image. (**c**) and (**d**) share the same scale bar.

MPicker was also tested on flattening the cylindrical and spherical surfaces (Supplementary Fig. 4). For cylindrical surfaces, a cylindrical influenza virion[13] was projected onto an elliptic cylindrical surface and flattened (Supplementary Fig. 4a). For spherical surfaces, MPicker provides two methods to flatten them. One treats spherical membranes as cylindrical surfaces and hence sacrifices their bottom and top parts that are largely smeared out by the missing-wedge effect, as demonstrated by the human immunodeficiency virus[14] (HIV) and Tick-borne encephalitis virus[15] (TBEV) (Supplementary Fig. 4b, c). The second method is to represent a surface in a triangle mesh (as in Supplementary Fig. 3) including the bottom and top parts, and flatten the whole sphere, as shown in Supplementary Fig. 4d.

## Visualizing macromolecules near the membrane

The flattened tomogram generated in MPicker provides enriched insight into not only protein organization on the membrane but also the relationship among macromolecules near the membrane. Several examples are presented below to demonstrate the advantages of flattened tomograms.

In red algae, the PBS and PSII extra-membrane domains are located on opposite sides of the thylakoid membrane. The PBS is on the stromal side, and the PSII extra-membrane domain is on the luminal side. By generating a flattened tomogram of a thylakoid membrane piece[3] (Fig. 4a), we examined the relationship between PBS and PSII. Previous studies have shown that PBSs are organized along parallel strings formed by PSIIs[3]. This feature was clearly visualized in the two slices of the flattened tomogram containing PBS and PSII (Fig. 4b). Furthermore, the PSII lines were disrupted in some regions (indicated by yellow arrows), which is consistent with the results of a previous report[16].

Thylakoid membranes are often stacked in a pairing manner, and PSII may play a role in this pairing[17]. We selected a pair of thylakoid membranes from *Chlamydomonas* (Fig. 4c) and generated a flattened tomogram (Fig. 4d). The two adjacent membranes were tightly appressed in a nearly parallel manner so that they could be flattened together. Two slices with PSII were extracted from the two membranes and were marked in green and magenta, respectively. Their spatial relationships on the membrane surface were visualized by merging the two slices. No significant overlap was observed between the PSII from adjacent membranes, which is consistent with the results of a previous study[2].

## Particle picking on the membrane

By flattening the membrane, particle picking in a 3D density map is simplified and becomes a task for 2D images. To facilitate particle picking, MPicker calculates and stores the coordinate mapping between the flattened and raw tomograms in order to convert the particle coordinates to each other.

Particle identification and selection depended mainly on the density of the extramembrane region in the tests. This is because the transmembrane region is often obscured by strong signals from the phospholipid bilayers. The structural features of a membrane protein, such as shape and contrast, often vary across slices in a flattened tomogram. Therefore, slices with obvious particle features in a flattened tomogram are a good choice for particle picking.

Particle picking in a flattened tomogram is similar to that used in single particle analysis (SPA). In addition to manual picking, MPicker wraps EPicker[18], a deep-learning-based software for automated particle picking on SPA micrographs. EPicker picks and assigns scores to particles in all (or user-specified) slices of a flattened tomogram. Since a particle in a flattened tomogram usually appears in multiple slices, MPicker removes duplicate picking based on scores, ensuring that the particle is picked only once.

Particle picking was evaluated on paired thylakoid membranes from four *Chlamydomonas* tomograms (from the same dataset shown in Fig. 4c). First, 559 particles with distinct features were manually picked from 20 flattened tomograms and used to train a deep-learning model of EPicker. EPicker was then used for automated particle picking

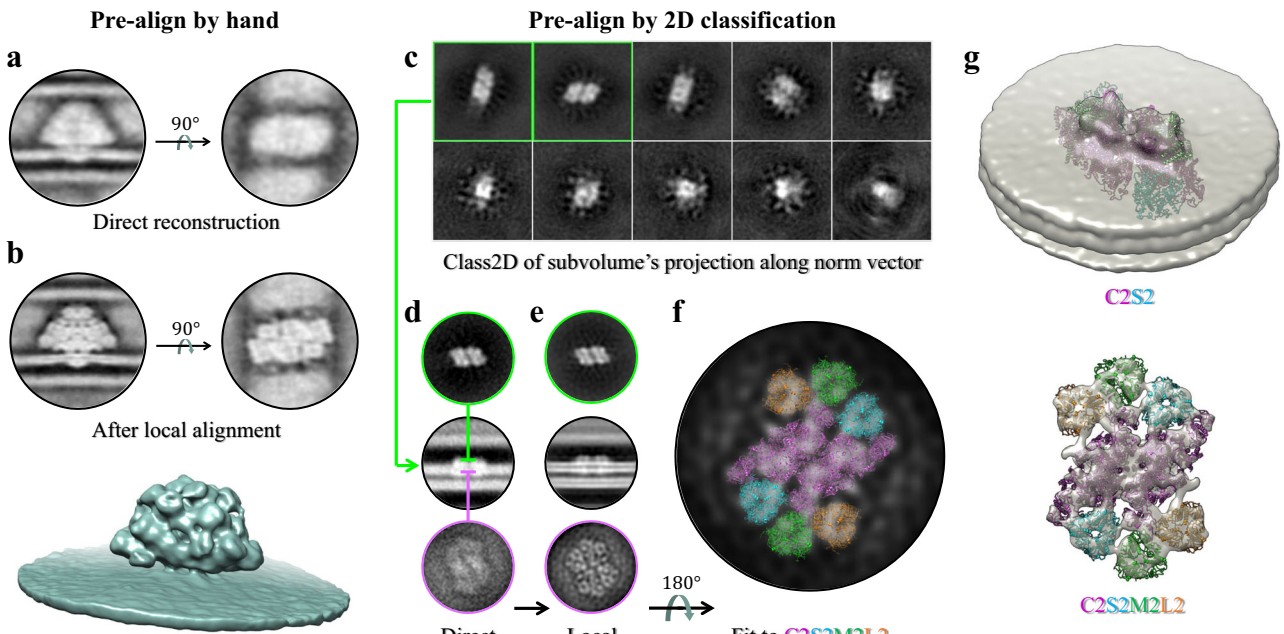

**Fig. 5 | Sub-tomogram averaging (STA) using orientations measured by MPicker. a** Representative section views of the red alga PBS density map reconstructed directly using the orientations manually labeled in MPicker. **b** Improved density map after local alignment. **c** 2D class averages of particles picked on *Chlamydomonas* thylakoid membranes. Two classes (green) with clear PSII features were selected for STA. **d** Section views of the PSII density map reconstructed directly using the orientations converted from 2D classification. Two $x–y$ slices (green, and magenta) are shown. Their positions in the $x–z$ slice (black) are marked by horizontal lines. **e** Section views of the improved density map after local alignment. **f** The model of PSII–LHCII supercomplex C2S2M2L2 (PDB: 6KAD) fitted to the map in (**e**). The components C, S, M, and L are drawn in different colors. **g** Density map after applying "Tomo CTF refinement" and "Tomo frame alignment" in RELION4. The map was shown at a low contour level of 0.3 and a high contour level of 1.95 for the extramembrane region and transmembrane region, respectively.

(Supplementary Fig. 5). To minimize false positives, we limited the particle search range to slices proximal to the thylakoid membrane surface, which roughly covered the extramembrane regions of membrane proteins. Initially, 5798 particles were picked from 192 flattened tomograms. After manual corrections, 1616 wrong picks were deleted and 1144 missed picks were added; thus, 5326 particles were finally used for subsequent analysis (see the following section). In practice, when combined with automated picking, the labor required for particle picking can be significantly reduced.

## Orientation estimation of membrane proteins

During membrane flattening, MPicker records the orientation (normal vector) of each membrane region. Membrane protein orientation is highly constrained by their interaction with the membrane and often limited to a 2D in-plane rotation on the membrane surface. Once the 2D orientation (in-plane rotation angle) is determined and combined with the membrane orientation, the 3D orientation of membrane protein particles can be deduced, allowing for direct 3D reconstruction. Considering the flexibility of proteins and possible flattening errors, further local alignment is often required to obtain the final reconstruction. This means that we can skip the global alignment in the STA and proceed directly to the local alignment, which helps reduce computational complexity. To determine the 2D orientation of a protein on a membrane, we propose two approaches. One is to manually label particle orientation on the flattened tomogram slices based on their shape characteristics, and the other is to utilize 2D classification. Two examples were used to demonstrate the two approaches.

For manual labeling, we chose red algae PBSs as an example, which showed a clear orientation signature in the flattened tomogram slice. A total of 268 PBS particles were picked from three red algae tomograms. Their 2D orientations were manually labeled in MPicker (like red arrows in Fig. 4b) and used to calculate the 3D orientation parameters and to directly reconstruct a density map using RELION2[19]

(Fig. 5a). The reconstruction was subsequently improved by further local alignment and achieved a resolution of 38.6 Å (Fig. 5b). The PBS rods are clearly shown in the map.

For the 2D classification approach, the particle projections along the normal vector of the membrane are subjected to alignment in 2D classification to derive the in-plane rotation angle on the membrane surface for each particle. Using the projection of an entire particle is not necessary, and projecting parts of the particle, such as the extramembrane region, is often a good choice. MPicker generates projections directly from raw tomograms to avoid the possible distortion and interpolation errors induced by flattening. Furthermore, because the cryoET data suffer from the missing wedge issue, the particle projection along the membrane normal vector may also be affected by the missing wedge, so the missing wedge must be considered during the 2D classification (Supplementary Fig. 6). We refer to RELION's 3DCTF model[19], which is used to process the contrast transfer function (CTF) and missing wedge in the STA. For each particle, we generated a 2D weighting image based on the 3DCTF model and the particle projection direction (see "Methods"), referred to as 2DCTF (Supplementary Fig. 6c). The 2DCTF integrates the information of both the missing wedge and the CTF. By combining the projection and corresponding 2DCTF of all the particles, we performed a 2D classification and determined the 2D orientations of the particles on the membrane.

We tested the above approach using 5326 particles picked in the previous section from *Chlamydomonas* thylakoids. MPicker was used to extract particle projections from the raw tomograms and compute the corresponding 2DCTF images. The 2D classification was performed using the THUNDER2[20] software (Fig. 5c). A total of 1846 particles from two classes (green boxes in Fig. 5c) with clear PSII structural features were selected, and the corresponding sub-tomograms were extracted from the raw tomograms for subsequent STA. The 3D orientation parameters of these PSII particles were calculated based on the 2D alignment results and the normal vectors of the corresponding

membrane region, and then were used for direct 3D reconstruction (Fig. 5d). After further local refinement, the reconstruction was improved, resulting in a map at a resolution of 25.4 Å (Fig. 5e, f). Then the "Tomo CTF refinement" and "Tomo frame alignment" in RELION4[21] were applied and finally improved the resolution to 24.2 Å (Fig. 5g, Supplementary Fig. 7a, b). The density map shows fine structural details matching previously reported in situ structures resolved by STA[2,22]. Interestingly, although the transmembrane region was omitted during the 2D classification, STA for the entire protein complex revealed a transmembrane region with distinct features. Upon fitting a SPA-derived *Chlamydomonas* PSII–LHCII supercomplex C2S2M2L2 model[23] (PDB entry code: 6KAD) to the resulting density map, the transmembrane region density showed high agreement with the model (Fig. 5f, Supplementary Fig. 7a). Some densities were also observed on the stromal side, which may be a region of a previously reported unidentified stromal protein (USP)[23] (red arrow in Supplementary Fig. 7b).

For the same PSII dataset, when calculating the particle orientations from scratch, we failed to obtain a reasonable result during the global alignment phase of STA. We then tried different alignment settings in RELION2 and finally achieved a more reasonable outcome by reducing the initial angular sampling step to 1.8°, as opposed to the previously used default value of 7.5°. Further analysis (see "Methods") indicated that the reduction in step size improved the in-plane angular alignment on the membrane surface (Supplementary Fig. 8a, b). Although the STA from scratch eventually got success, the quality of the reconstructed density maps did not match those obtained from the coarse alignment provided by MPicker, indicating that the alignment from scratch is not as good as that from MPicker (Supplementary Fig. 8c). We speculate that the membrane signal was excluded in the 2D classification of particles with only the extramembrane region, and hence a better estimation of in-plane rotation angles could be achieved. In the conventional 3D method, the membrane signal may inevitably influence the alignment, especially for the protein with a small and thin extramembrane region such as PSII.

In conclusion, the prior knowledge of in-plane rotation angle is useful to improve the robustness of the STA, as well as its computational efficiency, by skipping the challenging global alignment in the STA. Even if the 5798 particles selected by EPicker were used directly without manual correction, a similar reconstruction result could also be obtained following the same procedure (Supplementary Fig. 7c, d).

## Discussion

Identifying and analyzing membrane proteins from cryoET data is often challenging. We developed MPicker that facilitates membrane protein visualization and analysis by flattening membrane structures within the tomogram. The flattened tomogram provides a planar view that displays comprehensive protein information on the membrane in the form of slices, thereby simplifying the observation of protein distribution on the membrane surface. The flattened tomogram includes not only the membrane but also the adjacent densities on both sides of the membrane, which simplifies the analysis of the spatial relationship among macromolecules on both sides of the membrane. The flattened tomogram also simplifies protein localization on the membrane and orientation determination. The coordinates and orientations can be converted between flattened and raw tomograms, which can benefit STA calculation.

MPicker provides additional tools to handle complex membrane structures and low-quality data. First, MPicker supports the manual labeling of membrane surfaces when a membrane mask is missing. Second, MPicker supports a sophisticated method that uses a triangle mesh to define the shapes of complex membrane surfaces. Third, MPicker provides tools to avoid the misinterpretation of flattened membrane structures, including a tool to quantify the flattening-induced distortion and a 3D rendering tool to show undistorted membrane surfaces in 3D space.

Membrane flattening reduces the spatial freedom of membrane proteins from the 3D space to the 2D plane, which is beneficial for STA in at least two steps. First, flattening simplifies 3D particle picking in a tomogram to a 2D process similar to particle picking in SPA. Therefore, particle-picking tools designed for SPA, such as EPicker, can be applied to the flattened tomograms. Theoretically, flattened tomograms can also be processed using other particle-picking software designed for SPA. Second, MPicker assists global orientation estimation through manual labeling or 2D classification, making STA more computationally efficient than that directly from scratch. These features make MPicker serve as a more complete tool with more flexibility for analyzing membrane proteins than other tools (Supplementary Table 1).

Currently, membrane extraction requires manual interaction and is the most time-consuming part of MPicker. Automation is still needed, so that users can integrate the membrane flattening to their workflows without much manual intervention. Moreover, improved particle-picking methods that can take full advantage of flattened tomograms have yet to be developed. In addition, improving the 2D classification of protein projections on the membrane surface may be useful, which should be more computationally efficient than 3D classification in isolating small or unknown proteins from a mess of mixed molecules on the membrane. On the other hand, MPicker has been specifically optimized for particles on cellular membranes. In cases where membrane proteins are symmetrically or densely packed on viruses and tubular structures, particle-picking methods that rely on symmetry or oversampling are frequently employed. These methods may provide higher efficiency in particle picking compared to MPicker. However, MPicker's capability to flatten vesicles and tubes is a distinct advantage that sets it apart.

In summary, MPicker is a powerful tool to address the challenges of visualizing and analyzing membranes and their associated proteins for cryoET. MPicker was developed in Python, has an interactive graphical user interface (GUI) (Supplementary Fig. 9), and can be downloaded from https://thuem.net.

## Methods

### Surface extraction using membrane mask

The first step of membrane flattening in MPicker is to acquire a set of points on the membrane surface as starting points for further surface fitting. These starting points can be acquired semi-automatically based on a given membrane mask. To accomplish this, MPicker requires the user to first specify a few starting points on the membrane surface as seeds, termed seed points, and then extend these seed points to extract additional starting points. MPicker requires at least one seed point; more seed points can provide a more detailed selection of the membrane surface. When multiple seed points are specified, MPicker combines the extraction results of each seed point. For each seed point, the user needs to specify which one of the three axes, $X$-, $Y$-, or $Z$-axis, is the most perpendicular to the membrane surface. In the following, we describe how to extract the starting points from a mask based on one seed point in the case that the membrane surface is roughly perpendicular to the $X$-axis. For cases perpendicular to the other two axes, the algorithm can be achieved by switching the order of axes. The case perpendicular to the $Y$-axis is achieved by replacing $X$, $Y$, and $Z$ with $Y$, $X$, and $Z$, respectively. The case perpendicular to the $Z$-axis is achieved by replacing $X$, $Y$, and $Z$ with $Z$, $Y$, and $X$, respectively.

The process of extracting starting points from the mask can be divided into three steps. In the first step, MPicker extracts a boundary with a single voxel thickness from the mask (Supplementary Fig. 10a, b). MPicker provides two methods for boundary extraction: binary erosion on the mask (set the eroded points as the boundary), and layer-wise 2D Canny edge detection[24]. In the second step, MPicker extends the starting points from the seed point along the boundary. The extension is carried out within the $x$–$z$ plane by pixel-wisely increasing and decreasing $Z$ coordinates (blue curve in Supplementary

Fig. 10c). During the extension process, the $Z$ coordinates of the newly added points should change monotonically, otherwise, the extension is terminated. Users can also specify the maximum extension length to terminate the extension. In the third step, based on the points obtained from the previous step, MPicker further extended from each point (blue point in Supplementary Fig. 10c) along the boundary in a similar manner as the previous step but within the $x$–$y$ plane (magenta curve in Supplementary Fig. 10c). During the extension process, the $Y$ coordinates should change monotonically; otherwise, the extension is terminated. Finally, all the extracted points were combined to form the final starting points. MPicker separately applied one-dimensional extensions to the $x$–$z$ and $x$–$y$ slices to ensure that the surface formed by the extracted starting points was not too complex.

In practice, MPicker also removes outliers from the starting points. To determine the outliers, MPicker first calculates the average distance between each starting point and its 50 nearest neighbors and then calculates the mean and standard deviation of these average distances. Finally, the starting points whose average distance is greater than the mean plus two standard deviations are considered outliers. This method is commonly used in point-cloud processing[25].

### Generation of flattened tomogram

During the flattening, we defined three Cartesian coordinate systems for different usage. The Cartesian coordinate system consisting of the raw tomogram axes is denoted as $X$–$Y$–$Z$. For each membrane to be flattened, MPicker performs a translational and rotational transformation on the $X$–$Y$–$Z$ coordinate system to obtain a transitional Cartesian coordinate system, denoted as $x$–$y$–$z$. In the $x$–$y$–$z$ coordinate system, the membrane surface will be described in the form of a single-valued function $z = f(x,y)$ to simplify subsequent processing. The Cartesian coordinate system consisting of the flattened tomogram axes is denoted as $u$–$v$–$w$. For consistency with the terminology used in tomogram processing, the $X$–$Y$ slice of the raw tomogram and the $u$–$v$ slice of the flattened tomogram are referred to as the $x$–$y$ slice in this study. The "$x$–$y$" is unrelated to the $x$–$y$–$z$ coordinate system described here.

To obtain the coordinate system $x$–$y$–$z$ and describe the membrane surface in the form $z = f(x,y)$, MPicker needs to find a viewing direction as the $z$-axis, from which the membrane appears flat. To determine this direction, MPicker implements two methods. The first method involves fitting a plane to the starting points and using the direction perpendicular to the fitted plane as the $z$-axis. The second method is to find a projection direction that maximizes the projected area of the starting points and use it as the $z$-axis.

Once the $x$–$y$–$z$ coordinate system is determined, MPicker uses surface fitting to find the $f(x,y)$ that best matches the starting points. MPicker provides two surface fitting methods. The first method is polynomial fitting, which approximates the starting points with a $N$-th degree bivariate polynomial:

$$f(x,y) = \sum_{m=0,n=0}^{m+n \leq N} a_{mn} x^m y^n, \qquad (1)$$

where $N$ is specified by the user. The second method uses TPS interpolation (implemented using scipy.interpolate.RBFInterpolator). Compared to polynomial fitting, this method can better describe surfaces with complex shapes but requires more computations. Therefore, MPicker does not directly use all starting points (Supplementary Fig. 10d), but first downsamples these points (Supplementary Fig. 10e) to reduce the computation. MPicker then used these downsampled points as interpolation points and generated a continuous surface, $z = f(x,y)$, using TPS interpolation (Supplementary Fig. 10f). The users can specify the smoothness of the fitted surface: a higher smoothness may lead to a higher error between the surface and interpolation points. However, this also means better resistance to noise in the data. In practice, MPicker also removes interpolation points with large errors as outliers and then performs another round of TPS interpolation to further increase its robustness to noise.

After generating the surface represented by $z = f(x,y)$, the next step is to flatten the regions on both sides of the generated surface. To obtain the density of each voxel in the flattened tomogram, the mapping between the $u$–$v$–$w$ and $X$–$Y$–$Z$ coordinate systems must be determined (Supplementary Fig. 10g) in order to get the density at the corresponding voxel position in the raw tomogram. Once the density of each point in the $u$–$v$–$w$ coordinate system was obtained, the final flattened tomogram was formed (Supplementary Fig. 10h).

To establish the mapping between the $u$–$v$–$w$ and $X$–$Y$–$Z$ coordinate systems from the surface $z = f(x,y)$, MPicker first attempted to map the surface on a plane, which was the central slice of the flattened tomogram. Specifically, MPicker described the surface $z = f(x,y)$ using the following parametric equations:

$$\begin{cases} X = X(u,v) \\ Y = Y(u,v) \\ Z = Z(u,v) \end{cases}. \qquad (2)$$

The relationship between the $x$–$y$–$z$ and $X$–$Y$–$Z$ coordinate systems can be calculated as follows:

$$(X, Y, Z)^T = \boldsymbol{R} \times (x, y, z)^T + \boldsymbol{t}, \qquad (3)$$

where $\boldsymbol{R}$ represents the rotation matrix, $T$ is the transpose operation, $\boldsymbol{t}$ is the translation vector, and $\times$ is the matrix multiplication.

For a clear description of the flattening process, we assume that Eq. 2 was already obtained for now (details are described below), which means that the corresponding raw tomogram coordinates ($X(u,v)$, $Y(u,v)$, $Z(u,v)$) are known for each point $(u,v)$ in the central slice of the flattened tomogram. To further compute the corresponding raw tomogram coordinates for each point on either side of the central slice, MPicker extends each point in the central slice along the normal vector (and the reverse direction) of the surface. The number of extended pixels was specified by the user. Assuming that the number of extended pixels is $w_t$, the total thickness of the flattened tomogram is $2w_t + 1$ pixels. Further assuming that the central slice is the $w_c$-th slice in the flattened tomogram ($w_c = w_t + 1$), and the (normalized) normal vector of each point on the surface is represented by ($NX(u,v)$, $NY(u,v)$, $NZ(u,v)$), the mapping between the $u$–$v$–$w$ and $X$–$Y$–$Z$ coordinate systems can be obtained as follows:

$$\begin{cases} X(u,v,w) = X(u,v) + (w - w_c)*NX(u,v) \\ Y(u,v,w) = Y(u,v) + (w - w_c)*NY(u,v) \\ Z(u,v,w) = Z(u,v) + (w - w_c)*NZ(u,v) \end{cases}, \; |w - w_c| \leq w_t. \qquad (4)$$

At this point, the corresponding raw tomogram coordinates can be calculated for each voxel in the flattened tomogram. By performing trilinear interpolation on the raw tomogram, the density of each voxel in the flattened tomogram can be obtained.

To obtain normal vectors ($NX(u,v)$, $NY(u,v)$, $NZ(u,v)$) from Eq. 2, MPicker calculates the (unnormalized) normal vector using the following equation:

$$\begin{cases} NX = \frac{\partial Y}{\partial u}\frac{\partial Z}{\partial v} - \frac{\partial Z}{\partial u}\frac{\partial Y}{\partial v} \\ NY = \frac{\partial Z}{\partial u}\frac{\partial X}{\partial v} - \frac{\partial X}{\partial u}\frac{\partial Z}{\partial v} \\ NZ = \frac{\partial X}{\partial u}\frac{\partial Y}{\partial v} - \frac{\partial Y}{\partial u}\frac{\partial X}{\partial v} \end{cases}. \qquad (5)$$

This is also equivalent to first computing the (unnormalized) normal vector represented in the $x$–$y$–$z$ coordinate system by $\left(-\frac{\partial f}{\partial x}, -\frac{\partial f}{\partial y}, 1\right)$ and then computing its representation in the $X$–$Y$–$Z$

coordinate system by rotation. In practice, MPicker uses numerical differentiation methods to compute partial derivatives.

Now we explain how to get Eq. 2, in other words, how the surface is mapped to the central slice of the flattened tomogram. Since the surface can be described by $z = f(x, y)$, and the relationship between the $x-y-z$ and $X-Y-Z$ coordinate systems is already known (Eq. 3). If the relationship between $(u, v)$ and $(x, y)$ can be expressed as

$$\begin{cases} u = u(x, y) \\ v = v(x, y) \end{cases},\qquad(6)$$

then Eq. 2 can be obtained by combining Eq. 3, and 6, and $z = f(x, y)$.

The simplest method of generating Eq. 6 is to directly set $u(x, y) = x$ and $v(x, y) = y$. This is equivalent to projecting the surface directly on a plane, along the $z$-axis. In practice, MPicker projects the surface on the best-fit cylindrical surface and then unfolds it to minimize distortion. We assume that this best-fit cylindrical surface can be described by an $N$-th degree polynomial $z = h(y)$, as follows:

$$z(x, y) = h(y) = \sum_{n=0}^{N} a_n y^n,\qquad(7)$$

where $N$ is specified by the user. This means that the cylindrical surface is only bent on the $y$-axis. In this case, MPicker establishes the relationship between $(u, v)$ and $(x, y)$ as follows:

$$\begin{cases} u(x, y) = u(x) = x \\ v(x, y) = v(y) = \int_{y_{min}}^{y} \sqrt{1 + \left(\frac{d}{dy} h(y')\right)^2} \, dy' \end{cases},\qquad(8)$$

where $y_{min}$ is the bound of the tomogram. In this way, the equidistant sampling of $v$ corresponds to equal arc-length sampling on the cylindrical surface, effectively creating a 2D coordinate system $u-v$ on the cylindrical surface. In practice, MPicker uses numerical methods to compute the integral in Eq. 8 and the inverse function of $v(y)$, i.e., $y(v)$. To achieve a better fit for the cylinder, MPicker simultaneously optimized Eq. 3 and Eq. 7 to find the cylinder (implemented via scipy.optimize.least_squares), instead of performing fitting in two separate steps. Furthermore, to store the membrane in a flattened tomogram with a small size, MPicker might set the cylindrical surface bending direction to a suitable direction, not necessarily to the $y$-axis as described above.

In addition to projecting on a polynomial cylindrical surface, MPicker supports projection on an elliptic cylindrical surface. Since an elliptical cylindrical surface cannot be described by a single-valued function like $z = f(x,y)$ in the Cartesian coordinate system, the projection process differs slightly. In the first step, MPicker fits an elliptical cylinder based on the starting points. In the second step, a coordinate system is created on the elliptical cylinder surface. In the third step, the starting points were transformed into this new coordinate system for surface fitting. In short, the $x-y-z$ Cartesian coordinate system is replaced by a new coordinate system based on an elliptical cylinder, and its relationship with the $X-Y-Z$ coordinate system no longer satisfies Eq. 3. The following steps are the same as before, and here, MPicker directly sets $u(x,y) = x$ and $v(x,y) = y$, because MPicker creates the $x-y-z$ coordinate system directly on the cylindrical surface so that the equal arc-length sampling on the cylindrical surface already corresponds to the equidistant sampling on the $x-y$ plane.

### Distortion measurement

The distortion caused by flattening can be described locally by a linear transformation, that is, circles are deformed into ellipses. This is a conclusion drawn from computer graphics[9], and we only briefly explain it here and describe the specific method used by MPicker to calculate the degree of distortion.

Each flattened tomogram slice corresponds to a surface in 3D space. Suppose that the mapping between the slice and the surface can be described by Eq. 2, where $(u, v)$ are the 2D coordinates of a point on the flattened tomogram slice, and $(X, Y, Z)$ are the coordinates of the corresponding point in the original 3D space. If a point on this slice undergoes a small displacement $(du, dv)$, and its corresponding displacement in 3D space is $(dX, dY, dZ)$. The relationship between $(du, dv)$ and $(dX, dY, dZ)$ can be described as a linear transformation as follows:

$$\begin{bmatrix} dX \\ dY \\ dZ \end{bmatrix} = \begin{bmatrix} \frac{\partial X}{\partial u} & \frac{\partial X}{\partial v} \\ \frac{\partial Y}{\partial u} & \frac{\partial Y}{\partial v} \\ \frac{\partial Z}{\partial u} & \frac{\partial Z}{\partial v} \end{bmatrix} \begin{bmatrix} du \\ dv \end{bmatrix}.\qquad(9)$$

This means a small circle on the slice will be approximately mapped to a small ellipse on the surface. We assume that the circle is formed by a set of points $(u_0 + du, v_0 + dv)$ that satisfy $(du)^2 + (dv)^2 = (1\,pixel)^2$, and the corresponding ellipse is formed by points $(X_0 + dX, Y_0 + dY, Z_0 + dZ)$. To obtain the semi-major axis and semi-minor axis of the ellipse, we calculate the maximum length and minimum length of the vector $(dX, dY, dZ)$. According to Eq. 9, the square of the length can be calculated as follows:

$$\begin{aligned} \begin{bmatrix} dX & dY & dZ \end{bmatrix} \begin{bmatrix} dX \\ dY \\ dZ \end{bmatrix} &= \begin{bmatrix} du & dv \end{bmatrix} \begin{bmatrix} E & F \\ F & G \end{bmatrix} \begin{bmatrix} du \\ dv \end{bmatrix} \\ &= \begin{bmatrix} du' & dv' \end{bmatrix} \begin{bmatrix} \sigma_1^2 & 0 \\ 0 & \sigma_2^2 \end{bmatrix} \begin{bmatrix} du' \\ dv' \end{bmatrix}, \end{aligned}\qquad(10)$$

$$E = \vec{r_u} \cdot \vec{r_u}, \; F = \vec{r_u} \cdot \vec{r_v}, \; G = \vec{r_v} \cdot \vec{r_v}.\qquad(11)$$

Here, $\vec{r_u} = (\frac{\partial X}{\partial u}, \frac{\partial Y}{\partial u}, \frac{\partial Z}{\partial u})$ and $\vec{r_v} = (\frac{\partial X}{\partial v}, \frac{\partial Y}{\partial v}, \frac{\partial Z}{\partial v})$ are the tangent vectors of the surface, and $\cdot$ represent the vector dot products. In Eq. 10, $\sigma_1^2$ and $\sigma_2^2$ ($\sigma_1 \geq \sigma_2 > 0$) are the two eigenvalues of the matrix $\begin{bmatrix} E & F \\ F & G \end{bmatrix}$. The two eigenvalues are just the squares of the maximum and minimum lengths of the vector $(dX, dY, dZ)$, because $(du')^2 + (dv')^2 = (du)^2 + (dv)^2 = (1\,pixel)^2$. Therefore, $\sigma_1$ and $\sigma_2$ in Eq. 10 are just the semi-major axis and semi-minor axis of the ellipse.

In practice, MPicker uses numerical differentiation to compute the tangent vectors $\vec{r_u}$ and $\vec{r_v}$. In short, for each point $(u_0, v_0)$ on the flattened tomogram slice, MPicker only needs the corresponding raw tomogram coordinates of four nearby points $(u_0 \pm 1, v_0 \pm 1)$ to calculate the distortion caused by flattening at that point.

### Flattening of a surface represented in a triangle mesh

MPicker can also flatten the membrane surfaces represented by triangle meshes. To achieve this, the triangle mesh must have an appropriate parameterization[9]. Specifically, MPicker requires each vertex $(X, Y, Z)$ of the triangle mesh to be associated with a corresponding 2D coordinate $(u, v)$, similar to Eq. 2. Here $(u, v)$ are also referred to as the texture coordinates. Here, we illustrate the flattening process using the *Chlamydomonas* ER membrane as an example (Supplementary Fig. 3).

For user convenience, MPicker provides scripts to generate triangle meshes with texture coordinates. Users can use the starting points extracted from the mask using MPicker (or other forms of point clouds) as inputs (orange curve in Supplementary Fig. 3a). The mesh-generation process can be divided into three steps (Supplementary Fig. 3b). First, MPicker downsamples the starting points and removes outliers to obtain a point cloud with appropriate spacing. Second, MPicker reconstructs a triangle mesh from the point cloud using the Poisson surface reconstruction method[26] (provided in the library

Open3D[25]). Third, MPicker uses the OptCuts[11] software to compute the triangle mesh parameterization. In practice, the triangle mesh generated by surface reconstruction can be simplified by reducing the number of triangles to improve computational efficiency.

After obtaining a triangle mesh with appropriate texture coordinates, the flattening process was similar to that described in the previous section. Assuming the parameterization is appropriate, MPicker directly uses the texture coordinates $(u, v)$ as the $u$ and $v$ in Eq. 2. Since the vertex data are discrete, MPicker first uses the coordinates $(X, Y, Z)$ and the corresponding texture coordinates $(u, v)$ at each vertex to separately fit three continuous functions, $X(u,v)$, $Y(u,v)$, and $Z(u,v)$, using TPS interpolation. This gives Eq. 2. The remaining steps are the same as before: MPicker calculates the normal vector for each point using Eq. 5, and then determines the mapping between the $u$–$v$–$w$ coordinate system and the $X$–$Y$–$Z$ coordinate system using Eq. 4 and finally generates a flattened tomogram (Supplementary Fig. 3c). As before, MPicker could map the flattened tomogram slices back into 3D space (Supplementary Fig. 3d).

Here, MPicker uses OptCuts software and provides a wrapper for convenience. OptCuts is a software that can automatically calculate parameterization with small distortions for triangle meshes. The user simply specifies a threshold for the degree of distortion, and OptCuts minimizes the lengths of the cuts made on the meshes while ensuring that the average distortion remains below the specified threshold. The advantage of this method is that the surface can exhibit a small overall distortion after flattening, without being cut into fragments. OptCuts measures the distortion using $\sigma_1^2 + \sigma_2^2 + \sigma_1^{-2} + \sigma_2^{-2}$, where $\sigma_1$ and $\sigma_2$ have similar meanings as in Eq. 9.

### Preparation of tomograms

Most of the tomogram data used in this study were obtained from the Electron Microscopy Data Bank (EMDB), except for the *Chlamydomonas* tomogram data shown in Figs. 3a and 4c. In addition, we acquired and used the raw data of EMD-31243, EMD-31244, and EMD-31247 to calculate the STA of the red algae PBS[3].

The *Chlamydomonas* tomograms were prepared as follows. *Chlamydomonas reinhardtii* wild-type strain (mt-; CC-1691) used in this study is available from the *Chlamydomonas* Resource Center (University of Minnesota, St. Paul, MN, USA). Strains were cultured in Tris–acetate–phosphate (TAP) plates or liquid medium with aeration at $23 \pm 0.5\,°C$ with a light/dark cycle of 14/10 h at a light intensity of 8000 lx, as described previously[27]. When the cells reached an $OD_{600}$ of approximately 2, they were centrifuged at $751 \times g$ for 2 min 2–8 times to concentrate the cells. Then, 3 μL cell suspension was pipetted on glow-discharged grids (200 mesh Cu lacey carbon; glow-discharged using a PELCO easiGlow Glow Discharger, Ted Pella Inc), and 3 μL suspension buffer was added to the back of the grid. The grids were then blotted from the back and plunge-frozen using Leica EM GP (Leica Microsystems) set to 75% humidity, 25 °C temperature, and blotting time 6–8 s. The lamellae were thinned to approximately 150 nm using a dual-beam FIB-SEM system (Helios NanoLab DualBeam G3 UC, Thermo Fisher Scientific) with a cold stage (PP3010T, Quorum). CryoET data were collected using a 300 kV Titan Krios electron microscope (Thermo Fisher Scientific) equipped with a Cs corrector, BioQuantum energy filter, and K3 direct electron detector (Gatan). Micrographs were acquired in the super-resolution mode with 19,500× nominal magnification and 3.63 Å calibrated pixel size. Tilt series were acquired using SerialEM[28] with a bidirectional tilt scheme, ranging from 11° to 67° and then from 9° to −55°, with an angular increment of 2° and a defocus range of −4 to −6 μm. At each tilt angle, a micrograph consisting of 8 frames (0.25 s/frame) was collected, with a total dose of 124 $e^-/Å^2$ for the entire tilt series. The beam-induced motion was corrected using MotionCor2[29]. The defocus of the micrographs was determined using CTFFIND4[30]. Alignment and reconstruction of the tilt series were performed using the IMOD[4] software. Four tomograms abundant in thylakoid membrane pairs were selected for further processing. Tomograms binned by three folds were used for membrane segmentation, membrane flattening, and particle picking. Tomograms binned by two folds were used for 2D classification and STA.

### Preprocessing of tomograms

The contrast and SNR of a flattened tomogram depend directly on the contrast and SNR of the raw tomogram. Therefore, all tomograms used for flattening were preprocessed for better visualization using IsoNet[8] to denoise the tomograms and reduce the effect of missing wedge.

For the membrane mask, MPicker provides a built-in membrane segmentation functionality with a GUI. Membrane segmentation is based on a small pretrained neural network model of approximately 70 MB in size. The training data for this model were derived from five labeled public FIB-SEM datasets[31]. The data were converted into 72 sets of MRC files, each with a size of $200 × 200 × 200$ voxels, along with the corresponding labels for training. The training procedure is based on DeepTomo. The segmentation model used was based on the architecture of the 3D U-Net[32]. The feature extraction network consisted of a downsampling path and an upsampling path, each containing five blocks. Each block consisted of multiple convolutional layers, upsampling layers, instance normalization layers, and Leaky ReLU activation functions. For the training settings, we used the Adam optimizer with a gradually decaying learning rate in the order of $3 × 10^{-4}$, $10^{-4}$, $10^{-5}$, and $10^{-6}$. During training, we randomly cropped regions of $128 × 128 × 32$ voxels from individual tomogram data as input data using a batch size of eight and trained for 200 epochs. The training process was run on a single NVIDIA A100 GPU and took approximately 4 h.

Generation of a membrane mask using MPicker consists of two steps (Supplementary Fig. 1a). In the first step, a pretrained neural network is used to process the tomogram and generate continuous membrane scores. In the second step, the scored tomogram is postprocessed to obtain a final binary membrane mask. Post-processing includes Gaussian filtering of the scores, binarization based on a specified threshold, and noise removal using connected component analysis. Connected component analysis involves removing 2D small connected components in each $x$–$y$ slice and then removing 3D small connected components. For the tomogram in Fig. 2b ($960 × 928 × 178$ voxels), the segmentation took about 12 min on a workstation with two RTX 2080Ti GPUs.

### Particle picking by EPicker

MPicker wraps EPicker for particle picking in flattened tomograms and provides a GUI. Since EPicker is designed for 2D images, MPicker pre- and post-process the tomogram data. While generating the training dataset (Supplementary Fig. 5a), MPicker divides the flattened tomogram into several $x$–$y$ slices and converts the labeled 3D coordinates into the corresponding 2D coordinates on the slices. For particle picking, MPicker divides the flattened tomogram into several $x$–$y$ slices, uses EPicker to pick particles on each slice, and converts the 2D coordinates into 3D coordinates (Supplementary Fig. 5b). EPicker assigns a score to each candidate particle, and MPicker allows the user to specify the minimum score and maximum number of particles to be picked. Additionally, MPicker removes duplicates based on the spacing and scores of the particles to prevent the same particle from being picked multiple times on different slices. Since EPicker can only recognize 2D features, it is recommended to pick particles only in $x$–$y$ slices where the structural feature is clear.

To select membrane proteins on the flattened tomogram of the *Chlamydomonas* thylakoid membranes (Fig. 4d), 559 particles from 34 $x$–$y$ slices of 20 flattened tomograms were manually labeled for training. The trained model identified 5798 particles in 192 flattened tomograms. Since the extramembrane region of the protein is small, we usually picked particles on slices near the central slice, typically, in a range of three slices. Although the direct use of these particles could

yield the correct structure (Supplementary Fig. 7c, d), we manually corrected the picking results for better structural quality (Supplementary Fig. 5b) and obtained 5326 particles.

EPicker is a 2D image processing software, and its fast computational speed is one of its advantages. During training, we used a batch size of four and trained for 120 epochs, which took approximately 20 min. The particle-picking speed was approximately five slices per second. The calculations were performed using a single RTX 2080Ti GPU.

## 2D classification of projections with missing wedge

To automatically determine the 2D orientation of membrane proteins and screen high-quality particles, we performed 2D classification of the proteins selected from *Chlamydomonas* thylakoid membranes (Fig. 5c, Supplementary Fig. 7c). Since the normal vectors of the membrane at each point were obtained during flattening, MPicker projected each protein along its normal vector to obtain a 2D image. To preserve as much structural information as possible, MPicker extracts the projection images directly from the raw tomogram, which is binned by two folds with a voxel size of 7.26 Å. The resulting projection images were 70 × 70 pixels in size, with a projection depth of seven pixels.

For these projections, conventional 2D classification yielded poor results with noticeable stripe-like artifacts in the averaged 2D images (Supplementary Fig. 6a, b) possibly due to a misalignment caused by the missing wedge. According to the central slice theorem, a projection in real space corresponds to a slice in Fourier space. As 3D particles have a missing wedge in the frequency domain, the 2D images generated by projecting the particles also have a missing wedge in the frequency domain, the specific direction and size of which depend on the projection direction.

To reduce the effects of the missing wedge, weighting images describing both the CTF and missing wedge were used in the 2D classification (Supplementary Fig. 6c), with a function similar to that of the 3DCTF used in STA by RELION. To generate the weighting images, RELION was first used to generate a 3DCTF for each particle. MPicker then extracts a slice from the 3DCTF along the projection direction of each particle, according to the central slice theorem, to create a 2DCTF that serves as the weighting image of the particle. To use customized weights in 2D classification, THUNDER2, an upgrade of THUNDER[20] software, was used for classification. Without considering the missing wedge, the results obtained by THUNDER2 were similar to those obtained by RELION. However, when the effect of the missing wedge was considered, the performance of the 2D classification improved (Supplementary Fig. 6d), with clearer contours and the elimination of stripe-like artifacts. Regrettably, only PSII (1846/5326) was identified in the 2D averages. Improving the performance of 2D classification is a topic for future research. In practice, MPicker can generate a simple 3DCTF for each tomogram, allowing all the particles within the same tomogram to share a single 3DCTF, thereby saving time in generating the 3DCTF.

Compared with 3D classification, 2D classification offers advantages such as faster computation, higher stability, and independence from the initial models. In *Chlamydomonas* thylakoid membranes, the orientations of membrane proteins are generally consistent with the normal vectors, and the extramembrane regions are relatively flat, resulting in large projection areas that contain the major structural features of the proteins. During projection, we used only the density of the extramembrane region of the protein, because interference from membrane signals is often strong. Projections were used instead of slices to minimize the effect of coordinate errors in picking and allow for modeling the missing wedge using 2DCTF. Projecting subvolumes along the *z*-axis, followed by conventional 2D classification, is a common practice. This is a special case because projections along the *z*-axis do not have a missing wedge unless the tilt series does not contain the image at a 0° tilt angle.

## Calculation of Euler angles

Using the normal vectors of the membrane to constrain the angle search for membrane proteins in STA is a common practice; however, normal vectors can only provide partial constraints (two Euler angles out of three). Knowing the 2D orientation of the protein in the plane perpendicular to the normal vector is necessary to obtain complete protein orientation information and constrain the angle search. MPicker provides a function to manually label this orientation. In the flattened tomogram, after labeling the particle center, the user only needs to label another point to specify the direction (arrows in Fig. 4b). The line connecting these two points corresponds to the direction vector in the coordinate system of the raw tomogram. If this direction is taken as the *x*-axis of the local coordinate system of the particle, and the direction of the membrane normal vector is taken as the *z*-axis of the local coordinate system, then the complete orientation information can be calculated. Notably, the direction vector is usually not strictly perpendicular to the normal vector; therefore, in practice, MPicker uses its components perpendicular to the membrane normal vector.

Specifically, assuming that the normalized user-labeled direction vectors are $(vx_x, vx_y, vx_z)$ and the normalized normal vectors are $(n_x, n_y, n_z)$, the rotation matrix describing the particle orientation can be calculated directly as $\begin{bmatrix} vx_x & vy_x & n_x \\ vx_y & vy_y & n_y \\ vx_z & vy_z & n_z \end{bmatrix}$, where $(vy_x, vy_y, vy_z) = (n_x, n_y, n_z) \times (vx_x, vx_y, vx_z)$, and × represents the vector cross products. Orientation information in other forms, such as Euler angles, can also be converted directly from this rotation matrix.

Obtaining orientation information from 2D classification is similar to the process described above because the in-plane rotation angles obtained from 2D classification provide the same information as the direction vectors described above. MPicker provides scripts to convert the 2D classification results. In addition, MPicker can convert the 2D displacements calculated in the 2D classification into 3D displacements (in the plane perpendicular to the normal vector). MPicker also allows the user to specify additional translation and rotation angle for each class obtained from the 2D classification. This allows the merging of different classes in 2D classification, considering that the same type of protein may be classified into more than one class. The two selected classes in Fig. 5c were merged in this way before applying the STA.

## Calculation of sub-tomogram averaging

To calculate the STA of the red algae PBS, 268 PBS particles were manually selected from the three tomograms. The PBS particle orientations within the same tomogram were similar; therefore, three tomograms were used. Based on the manually labeled orientations in MPicker, RELION2 was then used for the direct reconstruction and local alignment. The tomograms used in the STA were binned by two folds to a voxel size of 6.86 Å. For direct reconstruction, RELION was set to perform 3D classification without alignment, and the number of classes was set to one. For local alignment, the result of the direct reconstruction was used as a reference, and "initial angular sampling" and "local searches from autosampling" in RELION were set to 7.5°. The C2 symmetry was also imposed, yielding a final resolution of 38.6 Å.

The STA workflow for *Chlamydomonas* PSII was similar to that for PBS. The tomograms used for the STA were binned by two folds to a voxel size of 7.26 Å. After 2D classification, the coordinates and orientations of 1846 particles from the four tomograms were used for direct reconstruction. For local alignment, the result of the direct reconstruction was used as a reference, and both "initial angular sampling" and "local searches from autosampling" in RELION were set to 3.7°. The C2 symmetry was imposed, yielding a map at 25.4 Å resolution. After alignment, "Tomo CTF refinement" and "Tomo frame alignment" in RELION4 were applied and improved the resolution to 24.2 Å. RELION4 was also used to reconstruct the final density map on unbinned data with a voxel size of 3.63 Å.

The final Fourier shell correlation (FSC) curves of the two STA result maps are shown in Supplementary Fig. 11.

## Sub-tomogram averaging of PSII without initial orientations

To compare the result of STA starting without initial particle orientation (obtained from 2D classification and membrane normal vectors by MPicker), we performed STA for PSII using RELION2 with the conventional workflow. The global alignment from scratch with the default initial angular sampling step of 7.5° failed. Then we reduced the step size to 3.7° and 1.8°, respectively, and performed STA with C2 symmetry imposed again. The results are shown in Supplementary Fig. 8.

The final orientations were compared to the results of our workflow (using the orientations estimated by MPicker as the initial orientations). We separated the orientations (three Euler angles) into two parts, normal vectors and in-plane angles. For the initial angular sampling step of 3.7° (Supplementary Fig. 8a), the differences in normal vectors were small (the median value was 0.6°), while the differences in in-plane angles were large (the median value was 33.8°). To investigate the difference, all particles were divided into two equally sized groups with a large (purple) and a small (orange) difference in the in-plane angles. The particles in the two groups were then 3D reconstructed, separately. For each group, the reconstruction was performed twice, one using the orientations obtained from the conventional workflow (3.7°), and the other using those obtained from our workflow (MPicker). Both the two reconstructions based on the orientations estimated by MPicker show clear structural features. However, the reconstructions based on the orientations obtained from the conventional refinement (3.7°) show a huge difference between the two groups, and the group with the large angular difference shows the wrong structural features of the complex, which indicates the misalignment for the in-plane rotation angle. For the initial angular sampling step of 1.8° (Supplementary Fig. 8b), similar analyses were carried out. The differences in normal vectors were still small (the median value was 0.5°), while the differences in in-plane angles were still large (the median value was 13.6°) but smaller than that of 3.7°. The reconstructions based on the orientations obtained from the conventional refinement (1.8°) also show a larger difference between the two groups, compared to that of our workflow (MPicker). This implied that the in-plane rotation angles obtained from our workflow were still more accurate because we assumed that the reconstructions of the two halves of particles should be similar if the orientation estimation was accurate. As for the final density maps obtained from different workflows (Supplementary Fig. 8c), our workflow gave the best result, and for the other two workflows starting from global alignments, a finer initial sampling step (1.8°) gave a better result.

## Computational efficiency

Membrane flattening is not a computationally intensive task, and hence MPicker requires just minimal computer performance. All tasks using MPicker in this work can be run on a typical laptop, except for the part of membrane segmentation and EPicker which require a general-purpose GPU. The time required for membrane flattening is typically several seconds, depending on the area of the membrane, the spacing of the interpolation points, and the surface projecting method (onto a plane or a cylindrical surface).

As an example, to generate a flattened tomogram in Fig. 4d, MPicker took ~3 s on surface extraction (5 seed points), ~8 s to generate a flattened tomogram (172 × 833 × 31 voxels, and flattening through a cylindrical surface and using 414 interpolation points with 12-pixels spacing). For another smaller surface (144 × 393 × 31 voxels for the flattened tomogram), MPicker took ~2 s on surface extraction (2 seed points) and ~3 s on flattening through a cylindrical surface (using 218 interpolation points with 12-pixel spacing). All these tests were run with an Intel Xeon Gold 6132 CPU.

## General software methods

MPicker is written in Python and uses several essential libraries, such as the NumPy, SciPy, Pillow, OpenCV, and Scikit-Image libraries for data processing; the mrcfile library for reading and writing MRC files; and the PyQt5 library for GUI development. The deep-learning framework used for membrane segmentation was based on the PyTorch library. The Open3D library was used for 3D rendering and geometry processing in MPicker. TPS interpolation in MPicker was implemented using the scipy.interpolate.RBFInterpolator. The cylindrical surface fitting uses scipy.optimize.least_squares as a nonlinear least-squares solver. Users will need to install EPicker, THUNDER2, and OptCuts individually if required, and MPicker provides wrappers for EPicker and OptCuts. RELION versions 2.1 and 4.0 were used for STA. We used the Open3D library and Chimera[33] for 3D visualization.

## Reporting summary

Further information on research design is available in the Nature Portfolio Reporting Summary linked to this article.

## Data availability

The entry codes of the tomograms from the Electron Microscopy Data Bank (EMDB) are EMD-10780, EMD-13771, EMD-11992, EMD-31243, EMD-31244, EMD-31247, EMD-10409, EMD-11075 and EMD-19003. The tomogram TS_43 in EMPIAR-10164 was also used. The entry codes of the structure models from the Protein Data Bank are 6KAD and 6KAC. The final density map of PSII–LHCII shown in Fig. 5g was deposited in the EMDB under accession number EMD-61019. The corresponding raw data was deposited in the EMPIAR under accession number EMPIAR-12469.

## Code availability

The program code is available for downloading from our project website [https://thuem.net] along with detailed information about software installation and usage. The code[34] is also available on GitHub [https://github.com/thuem/MPicker] and Zenodo [https://doi.org/10.5281/zenodo.14264179].

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

## Acknowledgements

This work was supported by funds from the National Natural Science Foundation of China (32241023 and 92254306 to X.L.), the Tsinghua-Peking Joint Center for Life Sciences, and the Beijing Frontier Research Center for Biological Structure. We acknowledge Zhengmao Wang of Junmin Pan lab for providing *Chlamydomonas* cells. We thank all the users for testing the software. We acknowledge the Tsinghua University Branch of China National Center for Protein Sciences Beijing for providing facility support in computing and cryoEM instruments.

## Author contributions

X.L. conceived the project; X.Y. designed and implemented the algorithm of membrane flattening; X.Y. wrote the major part of the program; S.L. developed the initial frame of graphic user interface; W.H. and N.Z. developed the membrane segmentation; H.W. prepared the cryoET data of *Chlamydomonas*; T.Z. supported the integration of EPicker; M.H. provided the support about THUNDER2; N.Z. and Y.S. offered useful suggestions; X.Y. performed the tests; X.L. and X.Y. wrote the manuscript; and all authors revised the manuscript.

## Competing interests

The authors declare no competing interests.
