## [Transparent Peer Review file · Nature Communications]

MPicker: Visualizing and Picking Membrane Proteins for Cryo-Electron Tomography

Corresponding Author: Dr Xueming Li

Version 0:

Reviewer comments:

Reviewer #1

(Remarks to the Author)

The manuscript "MPicker: Visualizing and Picking Membrane Proteins for Cryo-Electron 2 Tomography" by Yan and colleagues reports a software package for easing membrane visualizations and structural analysis of membrane proteins in electron cryotomograms. The workflow includes several steps: a (manual) segmentation of a part of the membrane, flattening of the membrane, and an account for distortions, visualization of molecules with possible picking, and sub-tomogram averaging. The package is tested on two molecules from the publicly accessible tomographic dataset of cryo-FIB-milled *Chlamydomonas* tomograms: PBS and PSII. MPicker is open source and has a nice-looking GUI.

I believe that the problem of particle picking is one of the important problems in sub-tomogram averaging and there is place for new tools. The promise of MPicker is sound, but I have two serious concerns about its utility.

The first major concern is that the set of nice tools provides very little conceptual novelty - manually prepicked membrane is triangulated, and straightened for visualization purposes. This has been done in the mentioned Membranorama (reference 2), however this tool has not been widely used (14 stars on Github). A similar triangulation approach is used in the Dynamo catalogue (paper from 2017) for regularization (DOI: 10.1016/j.jsb.2016.06.005) of the membrane for sub-tomogram picking. The software looks nice, however this report should be more detailed and published in a more specialized journal.

The second concern is that there is no benchmarking done in the manuscript. How good is the algorithm for drawing and triangulating membranes? Is it better than Membranorama or MemBrain (ref 7)? Would it work for full vesicles if they are not fib-milled? Will it work for small intracellular vesicles? Most importantly, the benchmarking datasets give structures of 25- 38 Angstrom, which is very low. Probably higher resolution can be achieved using the standard procedures. Can sub-megadalton proteins be processed like this? Could the authors use their software system to get a sub-nanometer structure of a moderately challenging protein, which can not be done by the other state-of-the-art methods? Maybe MPicker can do it faster and with fewer particles? None of these important benchmarks have been done.

A less important concern - it's hard to believe that it is impossible to align sub-tomograms of PSII, while membrane-embedded, it is very large. The authors used Relion2, for the other structures, perhaps they could use other packages or/and newer versions of Relion.

(Remarks on code availability)

Reviewer #2

(Remarks to the Author)

The manuscript by Yan et al. introduces MPicker, a novel software tool designed to perform the analysis of membrane proteins within cryo-electron tomography (cryoET) workflows. This work addresses a significant challenge in the field, such as the difficulty of observing and analyzing membrane proteins due to the inherent curvature and contrasting nature of

membranes within in situ tomograms.

The authors propose a membrane-flattening method implemented within MPicker. This innovative approach effectively reduces the spatial complexity of the analysis by generating a flattened tomogram. This significantly improves the visualization of proteins residing on and around membranes. The manuscript demonstrates the potential of MPicker for a more comprehensive, semi-automated workflow. This workflow encompasses segmentation of membranes, particle picking, and 2D classification of membrane proteins. This comprehensive approach, coupled with the successful application to diverse samples such as PSII, NPC, and PBS, highlights the potential of MPicker as a valuable tool for researchers utilizing cryoET.

One of the key strengths of this work lies in its ability to address a critical bottleneck in cryoET analysis. MPicker offers an elegant solution to the challenge of analyzing membrane proteins in tomograms, which are often obscured by the nature of membranes in tomograms. Furthermore, the potential for semi-automated segmentation, picking, and classification offers significant efficiency gains compared to traditional manual methods. The demonstration across various samples showcases the versatility of MPicker, suggesting its applicability to diverse cryoET workflows. The clarity of the manuscript's presentation makes it accessible to a broad audience within the field.

To further strengthen the manuscript, some suggestions for improvement are offered.

- Including data from highly curved membranes, such as viral particles (e.g. HIV, SarsCov2), vesicles, or membrane tubules, would strengthen the argument for MPicker's broader applicability. Additionally, providing more details on how MPicker handles drastic curvature changes, such as those observed in budding vesicles (Fig. 3(a-c)), would be beneficial.
- Highlighting the specific proteins mentioned in line 197 and their location within Fig. 3 would enhance clarity.
- Considering the recent release of RELION versions 4 and 5, a discussion on the potential benefits of re-running analyses with these newer versions or acknowledging the limitations of using RELION2 would strengthen the manuscript.
- A more detailed comparison between MPicker and existing software, emphasizing its advantages would bolster the case for its adoption.
- Briefly discussing potential future applications beyond the demonstrated examples, such as integration with downstream analysis pipelines, could further solidify MPicker's significance for the cryoET field.

Overall, this work by Yan et al. presents a promising software tool, MPicker, with the potential to significantly improve cryoET workflows for studying organelles and membrane proteins. Addressing the suggestions above would further strengthen the manuscript and solidify MPicker's impact on the field.

(Remarks on code availability)

Reviewer #3

(Remarks to the Author)

The authors present a method for visualization and localization of membrane proteins in cryo-electron tomograms. The core of the technique lies in membrane flattening which is very well described including its pitfalls in the form of possible distortions. In general, the authors describe all the essential steps in the workflow in a very clear manner and are critical of their shortcomings, even providing quantification for their uncertainty in some cases. This is rarely seen in recent method development/publications and the effort should be appreciated. The results show the robustness of their method w.r.t. the different cells as well as different membrane morphologies and orientations. The software provides a healthy mixture of automated and manual approaches for most of the steps. Despite many advances in cryo-electron tomography, particle picking is still not straightforward, and thus having tools that allow for easy manual corrections of automated approaches is often crucial for successful data processing. I also appreciate the possibility of using external inputs, e.g. membrane segmentation. MPicker has the potential to become a valuable tool for membrane proteins processing. I recommend the manuscript for publication provided the authors address the following points:

Major issue:

I could not find the STA maps produced within this study in the provided material. This is not acceptable - the maps need to be provided at least to the reviewers and ideally uploaded to the EMDB and released upon publication (in case they are connected to other unpublished manuscripts they can be released later but their accession code should be provided here already). The sentence about availability upon "reasonable request" cannot be meant seriously. On a similar note, the achieved resolutions are not supported by FSC curves, authors should add them.

Minor issues (in random order):

1) The functionality provided by Membranorama seems very similar to MPicker. Authors should provide more information on the differences between these two and possibly also the advantages of MPicker over Membranorama (if any). Adding a comparison on one tomogram might be interesting as well.

2) Each membrane present in a tomogram is flattened separately (I assume) and the automatic picking using EPicker is performed on the slices of flattened tomograms. Given that, the 1616 false positives out of 5798 picks and 1144 missing particles is rather poor performance.

Do the missing picks and false positives correlate to the regions of larger distortions? Or is it rather due to the 2D slices having different properties than typical raw 2D micrographs?

Was some other particle picking approach tried as well?

Is there support to upload coordinates from other approaches that would correspond either to the 2D slices or even 3D coordinates from raw tomograms (e.g. from 3D template matching on raw tomograms)?

3) Does Relion2 mean Relion version 2? If so, why use so old version?

4) Starting on line 293: "In the raw tomogram, the direction of the missing wedge was identical for all the particles (3D). For particle projections (2D) along the normal vectors of the membrane, the corresponding missing wedge shown in each projection differed among particles." This is misleading - the direction of the missing wedge is identical in cut-out subtomograms but after they are oriented (i.e. in correspondence to the normals) and then the direction differs. I suggest to rephrase this.

5) It is not obvious in the main text whether the 2D classification is done on raw or flattened tomograms - it becomes clear in methods, but I suggest stressing it also in the main text.

6) On line 356, "making STA more robust" is not a proper terminology in this case - robust STA would provide the structure without imposing a strong initial orientation. I would leave that out and keep only the statement on efficiency.

7) In the methods section authors mention in a few places that the results were computed on 2x downsampled/binning data. Why not bin even more? The membrane segmentation tends to work better with more binning due to better contrast and less noise (the segmentations are smoother). Would more binning reduce the need to simplify/downsample the point clouds/meshes later on?

8) Would be helpful to add some time estimates for the individual automated procedures in relation to the data size. This is mostly missing in the methods. Same for memory requirements and computational power needed for MPicker in general.

9) Are NX and NZ swapped in the equation s5 (line 512)? Normals are computed as the cross product of tangent vectors to the surface, no?

10) In equation s8 on line 533, I assume that the authors mean the arc length up to y , in which case the bounds could be added from $-\infty$ to y .

11) For the tilt-series acquisition - how many tilt-series were collected and how many were used for processing? Two small remarks on acquisition parameters - the defocus of -4 to $-6\mu\text{m}$ seems very high given the thin lamellae; the bidirectional acquisition scheme is not optimal for STA.

12) IsoNet was used for all preprocessing. To my knowledge this requires STA to be run to estimate the missing wedge. This might create a bit of a chicken-egg problem. Authors should comment more on how exactly the IsoNet was used. For some membranes, the missing wedge is not a problem, in such cases some denoising methods might be sufficient - was anything else than IsoNet tried? How does MPicker perform on non-denoised tomograms, e.g. those produced with the SIRT filter in IMOD?

13) Would the method benefit from 3D-CTF corrections on tomograms, or at least 2D-CTF correction on tilt-series?

14) On line 659 authors mention "image quality of tomograms". The quality of tomograms is ill-defined - the quality depends on the purpose. In this case, authors mean high contrast and less noise and they should define the term "image quality" as such before using it.

15) I am not familiar with DeepTomo (line 668) - is there possibly a reference missing there?

16) As for the calculation of Euler angles, the second point in manual labeling might be difficult to assign consistently even in the flattened tomograms and the issue can be even more complicated for complexes with C-symmetry. In practice, when surfaces are used to generate the normals of the particles, it is often enough to randomize the third missing (in-plane) angle and then to allow for full in-plane rotation during STA (not cone though). This procedure provides a sanity check for the quality of picking - if STA does not produce any reasonable average either the picking was incorrect or the data quality is very low. It would be good to add this feature of randomization as an option at this step. Also - what Euler angles are used (what convention)?

17) If I understood the webpage on MPicker correctly, the positions (and orientations) are outputted in a custom text format. Is there a possibility to export them directly as a starfile for Relion?

18) In Fig.3e: the lamina filaments are not visible and should not be pointed out. The same for the polysomes - their visibility is in general random and is not connected to the orientation of the membrane at all (and definitely not in the presented case). I suggest removing it from the legend as well as from the main text since it is misleading.

19) I suggest providing the tomogram in Fig.3d as a supplementary video.

20) In the extended data figure 1 - I suggest adding the corresponding raw tomogram next to b-d instead of referring to them in the legend.

(Remarks on code availability)

I have only checked the website and found it well done for its purpose. There is an installation guide and a tutorial one can follow. Currently, one has to download the software from the website - for the reproducibility and versioning (as well as other reasons) the authors should upload it to github or something similar. The installation should not be a problem since conda can be used and authors provide the complete environment for download as well. However, I have not tested it.

Version 1:

Reviewer comments:

Reviewer #1

(Remarks to the Author)

The revised MPicker manuscript has added several significant pieces of data, including the "unfolding" of tubular and spherical structures in the new Figure S9. These additions enhance the visualization of key findings, but in my view, the paper still faces major limitations.

First, there remains a lack of conceptual advance. The comparisons made to other software and the six arguments regarding implementation details represent only an incremental step forward. The authors have not addressed this issue in the revised manuscript, nor have they made any clear modifications to strengthen the description of their conceptual framework. If the authors believe there is a notable conceptual advancement, it would be highly beneficial to explicitly describe it in the text.

Additionally, I find one of the responses to my previous comments misleading. The authors state "...The signal from the membrane is very strong and makes alignment unreliable. This might be the reason that there is still no deposited in situ structure of PSII-LHCII in the EMDB." While this statement is technically not wrong, it is misleading because a structure of PSII alone was reported as early as 2010 (<https://doi.org/10.1016/j.bbabo.2010.11.007>), where particles clearly aligned. Although this study is old and the raw data is likely no longer available for benchmarking, it still needs to be cited.

A critical issue that remains is the lack of a quantitative evaluation of MPicker's performance for sub-volume averaging. This is one of the most interesting claims of the paper, yet there is no comparative data showing why MPicker outperforms conventional 3D methods for sub-volume averaging. The question is: "in which aspects is MPicker more beneficial for sub-volume averaging than the conventional 3D methods?" The PSII structure at 24 Å resolution is noteworthy, but the resolution is moderate, and no baseline data is provided for comparison. Given that the authors claim "automated particle picking and coarse alignment of membrane proteins for sub-tomogram averaging," it is important to demonstrate an advantage in this area—whether it's in processing speed, the number of particles picked, better alignment precision, or other relevant metrics. Using datasets such as HIV GAG, influenza virion (already present in the paper), or retromer on membranes (EMPIAR-10631), ion channels on membrane vesicles could help illustrate these benefits.

Minor note: The improved PSII structure should likely be moved into the main figure to provide better visibility within the manuscript.

(Remarks on code availability)

Reviewer #2

(Remarks to the Author)

The authors have addressed my comments and questions. Congratulations to all the authors on this great work.

(Remarks on code availability)

Reviewer #3

(Remarks to the Author)

The authors fully addressed all my questions and concerns, and all changes made to the manuscript further approved its readability. The authors showed the novelty of their approach and transparently compared it to the other existing software. From my perspective, if a method is novel, there is no need to outperform existing software to qualify for publication. Particle picking is an unresolved problem in cryo-electron tomography; any tool that can make this process easier, more accurate, or faster should be welcomed. I do not have any further concerns and can recommend the manuscript for publication.

I have a few remarks to the authors. However, their addressing is not necessary for the acceptance of the manuscript:

The response on line 499: I was not referring to the different environments between the in situ cellular samples and micrographs of purified proteins. I meant more different signal properties of the data. The 2D micrographs are essentially projections of whole structures, i.e., they contain a complete footprint of the structure. A slice from a tomogram has only

information from that slice (although potentially convoluted due to the missing wedge). This is a totally different type of information that is contained there. Therefore, I am not surprised that methods designed for particle picking on 2D micrographs might not perform as well for tomogram slices. This is also demonstrated by the good performance of the 2DCNN from EMAN2, which was trained exactly on the 2D slices, not projections. This may be something to consider for future development of pipelines using MPicker.

Regarding the response on the CTF correction: It is true that the resolution of the results gets affected by the lack of CTF correction only at subnanometer ranges. However, the shape of the structures is affected by the delocalization of the signal regardless of the resolution (nicely demonstrated in Fig. 3 in <https://doi.org/10.1016/j.jsb.2017.07.007>). For that reason, it might also be worth considering for MPicker.

The premise of MPicker is to provide user-friendly software, not just a method. Therefore, I would strongly recommend adding direct support for more coordinate formats. I found the argument that Relion2 was used because it has easier inputs and outputs funny. The authors were able to implement MPicker with advanced GUI; I am sure adding a few parsers for loading star files and corresponding writer functionality should not be problematic and would improve user experience. Installing one more software just to perform the conversion is annoying, and the chances of someone using Relion2 are very low.

Finally, in situ structures of phycobilisome and PSII are resolved at high resolution: <https://doi.org/10.1038/s41467-023-36175-y>.

(Remarks on code availability)

There were no changes in the code, so I stand by my previous statement.

Version 2:

Reviewer comments:

Reviewer #1

(Remarks to the Author)

The revised version of the M-picker manuscript went through minor improvements. It is a good technical manuscript and the software certainly can be useful for further investigations by the cryo-ET community.

Nonetheless, my major concern from the previous round of revisions has not been addressed "A critical issue that remains is the lack of a quantitative evaluation of MPicker's performance for sub-volume averaging. This is one of the most interesting claims of the paper, yet there is no comparative data showing why MPicker outperforms conventional 3D methods for sub-volume averaging. The question is: "in which aspects is MPicker more beneficial for sub-volume averaging than the conventional 3D methods?" The PSII structure at 24 Å resolution is noteworthy, but the resolution is moderate, and no baseline data is provided for comparison. Given that the authors claim "automated particle picking and coarse alignment of membrane proteins for sub-tomogram averaging," it is important to demonstrate an advantage in this area—whether it's in processing speed, the number of particles picked, better alignment precision, or other relevant metrics. Using datasets such as HIV GAG, influenza virion (already present in the paper), or retromer on membranes (EMPIAR-10631), ion channels on membrane vesicles could help illustrate these benefits"

If the authors are to publish the paper, they should deposit the tomograms and the positions of PSII particles that were used for an unsuccessful subtomogram alignment attempt with Relion2 (lines 344 onwards). EMPIAR would be a good platform for it.

Minor:

I would not call the extraction of a membrane to 2D to be a fundamental concept, as said in line 84 of the updated MS.

Line: 1244 : "This relies on the software THUNDER for the 2D classification". Each software will have the classification functionality if external software is used, probably all boxes should be ticked.

(Remarks on code availability)

Reviewer #2

(Remarks to the Author)

The authors have addressed my comments and questions and I think this manuscript should be published. Congratulations to all the authors.

(Remarks on code availability)

Reviewer #3

(Remarks to the Author)

As already stated before, from my perspective, the authors provided satisfactory answers, and the manuscript is ready to be published.

A final note to the authors: the difference between in situ and in vitro structure does not come from the averaging technique used but from the sample. Thus, the structure obtained by GISPA counts as an in situ structure of the phycobilisome because it was obtained from a crowded cellular environment, not from a purified sample.

(Remarks on code availability)

Commented on before.

We thank all reviewers for their valuable comments and suggestions. Comments from reviewers are
highlighted in grey background throughout the text.

Reviewer #1 (Remarks to the Author):

The manuscript “MPicker: Visualizing and Picking Membrane Proteins for Cryo-Electron
Tomography” by Yan and colleagues reports a software package for easing membrane visualizations
and structural analysis of membrane proteins in electron cryotomograms. The workflow includes
several steps: a (manual) segmentation of a part of the membrane, flattening of the membrane, and
an account for distortions, visualization of molecules with possible picking, and sub-tomogram
averaging. The package is tested on two molecules from the publicly accessible tomographic dataset
of cryo-FIB-milled Chlamydomonas tomograms: PBS and PSII. MPicker is open source and has a
nice-looking GUI.

I believe that the problem of particle picking is one of the important problems in sub-tomogram
averaging and there is place for new tools. The promise of MPicker is sound, but I have two serious
concerns about its utility.

The first major concern is that the set of nice tools provides very little conceptual novelty – manually
prepicked membrane is triangulated, and straightened for visualization purposes. This has been done
in the mentioned Membranorama (reference 2), however this tool has not been widely used (14 stars
on Github).

24 A: Our innovation lies in generating flattened tomograms which is the major difference from other
triangulation-based methods and accordingly brings more advantages in visualization and particle
picking. With MPicker, many previous challenges in membrane analysis become simple. We
provided a detailed comparison among software, MPicker, Membranorama, and Dynamo, and
summarized their differences in the following **Table R1**. This table has been added in the revised
manuscript as **Extended Table 1**.

**Table R1 | Comparison of functions among three software.**

	MPicker	Membranorama	Dynamo catalogue
Inputs	Membrane mask, or labeled points, or triangle mesh	Triangle mesh	Labeled points
Generate flattened tomogram	✓	✗	✗
Generate triangle mesh	✓	✗	✓
Generate normal vector	✓	✓	✓
Label in-plane angle	✓	✓	✗
Render 3D surface with grey level	✓	✓	✗
Particle picking	In flattened tomogram	On 3D surface	In raw tomogram, or oversampling

While Membranorama has some functional overlaps with MPicker, such as visualizing

membrane surfaces and manually picking particles, particle picking and analyzing on the membrane
was never an easy task before MPicker. MPicker serves as an efficient solution for the *in situ*
analysis of membrane proteins. Below, we list some practically essential differences between the
two software:

- 1) MPicker warps a sub-tomogram region based on the shape of the membrane surface,
generating a new flattened tomogram. Membranorama utilizes the raw tomogram to obtain
grey-level information on the membrane surface and renders it in 3D space. The key
advantage of MPicker is that there is more 3D information from the flattened tomogram than
from a simple 3D surface rendering.
- 2) MPicker uses a radial basis function (RBF) to represent the surface, and users only need to
provide a membrane mask or even manually annotated points. In contrast, Membranorama
requires users to provide a triangle mesh (such as an obj file) to represent the membrane
surface, which is not easy to obtain. MPicker also supports triangle mesh as the input.
- 3) By generating flattened tomograms, MPicker bypasses 3D rendering. Picking particles in
flattened tomograms is just like the particle picking in ordinary tools, accordingly, automated
picking becomes convenient. 3D rendering and triangulation also become simple on the basis
of the flattened tomogram, and are provided as an auxiliary function of MPicker.
Membranorama performs most operations (such as particle picking) in 3D, which makes the
operation in one more dimension, and hence not as straightforward and smooth as in 2D. And
only manual particle picking is supported in Membranorama.
- 4) MPicker can output a triangulated membrane surface (obj file), which carries the texture and
can be loaded by other software (such as the Windows 3D viewer or Blender, as well as
Membranorama).
- 5) MPicker generates a separate flattened tomogram for each membrane, allowing users to
process membranes one by one and easily visualize the relationship of adjacent membranes.
Moreover, flattened tomograms are ordinary MRC files, and users can use any favorite
software to pick particles or do other processing in the flattened tomograms. To our
knowledge, Membranorama doesn't well support processing individual membrane separately
in the case of many membranes stacked together (such as **Fig. 4c**).
- 6) MPicker can also correlate and display the point positions on the fly between the raw and
flattened tomograms. By providing multiple views of the same region users can easily locate
and identify proteins. In contrast, Membranorama displays only the surface of the membrane,
corresponding to the xy slice of the flattened tomogram in MPicker.

In summary, MPicker uses different an idea to reduce the difficulty in practical data analysis
and improve the efficiency. Moreover, the flattened tomograms provide the opportunity for users to
analyze the membrane structure in a customized way.

A similar triangulation approach is used in the Dynamo catalogue (paper from 2017) for
regularization (DOI: 10.1016/j.jsb.2016.06.005) of the membrane for sub-tomogram picking. The
software looks nice, however this report should be more detailed and published in a more specialized
journal.

76 A: Dynamo catalogue and MPicker focus on different problems. Dynamo can triangulate a
77 membrane surface based on manually annotated points, but it cannot visualize the membrane surface

or assist in membrane protein identification and localization, so users still need to obtain particle
coordinates using other methods. The purpose of triangulating the membrane in Dynamo is to obtain
the normal vector of the membrane and use it as a constraint in sub-tomogram averaging (STA).
MPicker has the similar function, and provides powerful functions, such as estimating in-plane angle
(as discussed in the section "**Orientation estimation of membrane proteins**"). Another function
of Dynamo is to oversample particles on the membrane, which can also be done in the flattened
tomogram.

The second concern is that there is no benchmarking done in the manuscript. How good is the
algorithm for drawing and triangulating membranes? Is it better than Membranorama or MemBrain
(ref 7)?

90 A: For the visualization purpose, MPicker supports the mesh triangulation and rendering (drawing),
and has a similar performance as Membranorama. However, the flattening proposed in MPicker
provides more advantages in visualization than the triangulation method. In fact, MPicker is
designed to check proteins in flattened tomograms, which is more convenient than in 3D. Displaying
the surface in 3D is just an optional auxiliary function in MPicker.

Regarding triangulation, both Membranorama and MemBrain require users to provide the
triangle mesh. MPicker does not require a triangle mesh, and can triangulate the membrane after
flattening, as a plane can be easily triangulated (e.g., dividing each square pixel into two triangles).

For rendering, once the grey level of each point on a surface is known, rendering is a well-
established technique. MPicker uses the Open3D library for rendering, with just minimal parameters
to be set. During the revision, a new function was added to output the membrane's triangle mesh
and corresponding texture so that users can operate them in their favorite software. To compare the
3D rendering effects, we exported the triangle mesh of a membrane using MPicker and rendered it
using Membranorama, as shown in **Fig. R1**. Both tools can meet the requirements for visualization
and particle picking.

**Fig. R1 | Comparison of membrane drawing between MPicker and Membranorama. a,** The
 raw tomogram (the same as **Fig. 1a**) and an x-y slice of flattened tomogram (orange). The surface
 corresponding to the slice was exported as a triangle mesh and used as the input of Membranorama.
 **b,** The same surface in **a** is presented in 3D by MPicker and Membranorama, which illustrates the
 similar display quality.

 **Would it work for full vesicles if they are not fib-milled? Will it work for small intracellular vesicles?**

115 A: Yes, MPicker can process vesicles from fib-milled or non-fib-milled samples. MPicker supports
 two methods to flatten spherical structures.

For the first method, by ignoring the top and bottom parts of the sphere (which are often fuzzy
 due to the missing wedge effect), MPicker treats the vesicle as a cylinder and projects the surface
 onto an elliptical cylinder for flattening. For the second method, MPicker utilizes the software
 OptCuts to flatten arbitrarily shaped surfaces with minimal distortion, as demonstrated in **Extended**
 **Data Fig. 3**. This method can also be applied to any spherical structures. In practice, the accuracy
 and smoothness of the membrane surface become critical for small objects.

We tested the two methods and showed the results by adding a new figure **Extended Data Fig.**
 **9** (a copy of this figure is shown below) in the revised manuscript and discussed in the section
 "**Membrane flattening on various membranes**". For the first method, we tested it on a cylindrical
 influenza virus (**Extended Data Fig. 9a**) and a spherical HIV virion (**Extended Data Fig. 9b**). For
 the second method, in order to avoid the interference of the missing-wedge effect, we chose a density
 map of tick-borne encephalitis virus calculated by SPA, considering it as a sphere with ~460 Å
 diameter, and flattened it with two different projecting strategies (**Extended Data Fig. 9c** and **d**).

**Extended Data Fig. 9 | Examples of the flattened cylindrical and spherical surface.** To illustrate
 the performance of MPicker on flattening cylindrical and spherical surfaces, we chose three viruses
 as examples. **a,** Flattening the outer surface of an influenza virion in a tomogram (EMD-11075) by

projecting it on a cylindrical surface. **b**, Flattening the outer surface of an HIV virus-like particle in
 a tomogram (TS_43 in EMPIAR-10164) by projecting it on a cylindrical surface. **c**, Flattening a
 density map (EMD-19003) of the TBEV virion by projecting it on a cylindrical surface. The
 membrane of TBEV was modeled as a sphere with the diameter of 46 nm. **d**, Flattening the same
 TBEV virion in **c** by triangle-mesh-based method. The processing workflow was the same as that
 used in **Extended Data Fig. 3**.

**Most importantly, the benchmarking datasets give structures of 25- 38 Angstrom, which is very low.**

145 A: MPicker is designed to assist in STA, such as particle picking and initial orientation estimation.
 But pushing STA to high/atomic resolution is not the task of MPicker and is not in the scope of this
 manuscript. Getting high resolution depends on many factors involving the data quality, number of
 particles, and algorithm of STA, which is not involved in this work. STA of membrane proteins in
 FIB-milled samples is still very challenging.

**Probably higher resolution can be achieved using the standard procedures.**

153 A: As we mentioned in the manuscript, for the same PSII dataset, when calculating the particle
 orientations from scratch by global alignment, we failed to obtain a reasonable reconstruction using
 STA. Following a standard procedure of STA, we tried many different settings but unfortunately,
 none of them worked, as shown in **Fig. R2**. Maybe, a more skillful expert can get better results than
 that we got. At least, these results demonstrated that MPicker makes a successful STA more
 obtainable.

**Fig. R2 | STA of PSII–LHCII using different workflows.** **a**, The same density map as shown in
**Fig. 5e**. Local orientation alignment of Relion2 was applied on particles selected from 2D
classification, which generated the best result. **b**, The density map generated in the same workflow
as **a** but using Relion4. The structure is not as clear as that in **a**. **c-e**, Some results of 3D refinement
using “general” STA workflows. The structure features are unclear. **f**, Results of 3D classification
using Relion2. The structure features are unclear. Two of the five classes are empty. **g**, Results of
3D classification using Relion4. The structure features are unclear. One of the five classes is empty.
In all the panels, the notions are described as follows. Images with red boxes are x–y slices crossing
the extramembrane region. Images with green boxes are x–z slices crossing the volume center.
Images with blue boxes are y–z slices crossing the volume center. Images with black boxes are x–y
slices crossing the transmembrane region. “Relion2” or “Relion4” means the program used for 3D
refinement or 2D classification. “part” means only 1846 particles selected from 2D classification
(**Fig. 5c**) were used. “all” means all 5326 picked particles were used. “local” means local orientation
alignment was carried out, based on the Euler angles converted from 2D classification. “global”
means global orientation alignment was carried out. “localnorm” means membrane normal vectors
were used to constrain the orientation alignment (by setting “--sigma_tilt 10” in Relion4). For
“local”, the initial reference was reconstructed directly using the Euler angles converted from 2D
classification. For “global” and “localnorm”, only normal vectors were used to reconstruct the initial
reference, the in-plane angles were set to random values.

Can sub-megadalton proteins be processed like this?

183 A: Yes. Theoretically, the effectiveness of MPicker depends on how many recognizable features are
184 visible in the flattened tomogram. In fact, the extramembrane region of PSII is just about 300 kD.
Anyway, resolving small membrane proteins is always challenging.

Could the authors use their software system to get a sub-nanometer structure of a moderately
challenging protein, which can not be done by the other state-of-the-art methods? Maybe MPicker
can do it faster and with fewer particles? None of these important benchmarks have been done.

191 A: As discussed before, many factors can influence the resolution of STA, which are not handled by
192 MPicker. On the other hand, for the PSII data tested in this work, the conventional procedures did
not yield the same or better results, as shown in **Fig. R2**.

A less important concern - it's hard to believe that it is impossible to align sub-tomograms of PSII,
while membrane-embedded, it is very large.

198 A: As mentioned above, conventional procedures indeed fail to yield satisfactory results. STA of
199 membrane proteins in FIB-milled samples is always challenging. For PSII, most protein regions are
200 embedded in the membrane. The signal from the membrane is very strong, and makes alignment
unreliable. This might be the reason that there is still no deposited in situ structure of PSII-LHCII
in the EMDB.

The authors used Relion2, for the other structures, perhaps they could use other packages or/and

newer versions of Relion.

207 A: The reason to use Relion2 is that its input/output data format is simpler than the newer version,
which simplifies the data conversion. We also tried to use Relion4 with various settings and found
that the results were not improved and sometimes worse, as shown in **Fig. R2**.

Nevertheless, we found that enabling the functions of “Tomo CTF refinement” and “Tomo
frame alignment” with fixed particle orientations in Relion4 improved the resulting map a little bit.
Consequently, we replaced the result in **Extended Data Fig. 6 a-b** with the new map and updated
the final resolution from 25.4 Å to 24.2 Å. We also updated the corresponding descriptions in the
sections "**Orientation estimation of membrane proteins**" and "**Calculation of subtomogram**
**averaging**" in the revised manuscript.

Reviewer #2 (Remarks to the Author):

The manuscript by Yan et al. introduces MPicker, a novel software tool designed to perform the
analysis of membrane proteins within cryo-electron tomography (cryoET) workflows. This work
addresses a significant challenge in the field, such as the difficulty of observing and analyzing
membrane proteins due to the inherent curvature and contrasting nature of membranes within in situ
tomograms.

The authors propose a membrane-flattening method implemented within MPicker. This innovative
approach effectively reduces the spatial complexity of the analysis by generating a flattened
tomogram. This significantly improves the visualization of proteins residing on and around
membranes. The manuscript demonstrates the potential of MPicker for a more comprehensive,
semi-automated workflow. This workflow encompasses segmentation of membranes, particle
picking, and 2D classification of membrane proteins. This comprehensive approach, coupled with
the successful application to diverse samples such as PSII, NPC, and PBS, highlights the potential
of MPicker as a valuable tool for researchers utilizing cryoET.

One of the key strengths of this work lies in its ability to address a critical bottleneck in cryoET
analysis. MPicker offers an elegant solution to the challenge of analyzing membrane proteins in
tomograms, which are often obscured by the nature of membranes in tomograms. Furthermore, the
potential for semi-automated segmentation, picking, and classification offers significant efficiency
gains compared to traditional manual methods. The demonstration across various samples
showcases the versatility of MPicker, suggesting its applicability to diverse cryoET workflows. The
clarity of the manuscript's presentation makes it accessible to a broad audience within the field.

To further strengthen the manuscript, some suggestions for improvement are offered.

-Including data from highly curved membranes, such as viral particles (e.g. HIV, SarsCov2), vesicles,
or membrane tubules, would strengthen the argument for MPicker's broader applicability.

248 A: Thanks for the constructive suggestions. To address this, we have included additional tests on
cylindrical influenza virus and spherical HIV in tomograms. We also used a density map spherical
TBEV virus from SPA, which isn't influenced by the missing-wedge effects and can demonstrate
the performance in flattening vesicles. The results were added as a new figure **Extended Data Fig.**
**9** (a copy is shown below) in the revised manuscript and discussed in the section "**Membrane**
**flattening on various membranes**". Additionally, **Extended Data Fig. 3** provides another example
of flattening a complicated surface.

Generally, flattening highly curved membranes is susceptible to the smoothness and resolution
of the surface segmentation. The actual performance may vary in different cases. In the future
version of MPicker, we are planning to enhance the capability of MPicker for highly curved
membranes.

Extended Data Fig. 9 | Examples of the flattened cylindrical and spherical surface. To illustrate the performance of MPicker on flattening cylindrical and spherical surfaces, we chose three viruses as examples. **a**, Flattening the outer surface of an influenza virion in a tomogram (EMD-11075) by projecting it on a cylindrical surface. **b**, Flattening the outer surface of an HIV virus-like particle in a tomogram (TS_43 in EMPIAR-10164) by projecting it on a cylindrical surface. **c**, Flattening a density map (EMD-19003) of the TBEV virion by projecting it on a cylindrical surface. The membrane of TBEV was modeled as a sphere with the diameter of 46 nm. **d**, Flattening the same TBEV virion in **c** by triangle-mesh-based method. The processing workflow was the same as that used in **Extended Data Fig. 3**.

Additionally, providing more details on how MPicker handles drastic curvature changes, such as those observed in budding vesicles (Fig. 3(a-c)), would be beneficial.

A: In the **Methods** section, we introduced the tool OptCuts to map the surface onto a plane (as in **Extended Data Fig. 3** and **Extended Data Fig. 9d**), which also be used to handle drastic curvature changes. The challenges of flattening curved surfaces (surface parameterization) remain an important topic in the field of computer graphics. OptCuts can automatically cut the surface and minimize distortions caused by flattening.

We tried to flatten the entire budding vesicle using OptCuts, as shown in **Fig. R3b**. However the results didn't show more molecular information on the flattened membrane. Possible reasons might be: inaccurate membrane localization because many regions of the membrane are not visible in the tomogram, weak membrane protein signals on the vesicle, too large curvature and the influence of the missing wedge effects. Anyway, flattening a highly curved surface is still very challenging.

**Fig. R3 | Flattening the budding vesicle.** The membrane of vesicle 1 in **Fig.3a** and the nearby
 membrane were used together for flattening (orange curve). An x-y slice in the flattened tomogram
 and the corresponding 3D surface were shown.

-Highlighting the specific proteins mentioned in line 197 and their location within Fig. 3 would
 enhance clarity.

298 A: Thanks for the suggestion. We updated the **Fig. 3c** in the revised manuscript.

-Considering the recent release of RELION versions 4 and 5, a discussion on the potential benefits
 of re-running analyses with these newer versions or acknowledging the limitations of using
 RELION2 would strengthen the manuscript.

304 A: The reason to use RELION2 is that its input/output data format is simpler than the newer version,
 which simplifies the data conversion. We also tried to use RELION4 with various settings and found
 that the results were not improved and sometimes worse, as shown in **Fig. R2**.

Nevertheless, we found that enabling the functions of “Tomo CTF refinement” and “Tomo
 frame alignment” with fixed particle orientations in RELION4 improved the resulting map a little
 bit. Accordingly, we replaced the result in **Extended Data Fig. 6 a-b** with the new map and updated
 the final resolution from 25.4 Å to 24.2 Å. We also updated the corresponding descriptions in the
 sections "**Orientation estimation of membrane proteins**" and "**Calculation of subtomogram
 averaging**" in the revised manuscript.

Fig. R2 | STA of PSII-LHCII using different workflows. **a**, The same density map as shown in **Fig. 5e**. Local orientation alignment of Relion2 was applied on particles selected from 2D classification, which generated the best result. **b**, The density map generated in the same workflow as **a** but using Relion4. The structure is not as clear as that in **a**. **c-e**, Some results of 3D refinement using “general” STA workflows. The structure features are unclear. **f**, Results of 3D classification using Relion2. The structure features are unclear. Two of the five classes are empty. **g**, Results of 3D classification using Relion4. The structure features are unclear. One of the five classes is empty. In all the panels, the notions are described as follows. Images with red boxes are x–y slices crossing the extramembrane region. Images with green boxes are x–z slices crossing the volume center. Images with blue boxes are y–z slices crossing the volume center. Images with black boxes are x–y slices crossing the transmembrane region. “Relion2” or “Relion4” means the program used for 3D refinement or 2D classification. “part” means only 1846 particles selected from 2D classification (**Fig. 5c**) were used. “all” means all 5326 picked particles were used. “local” means local orientation alignment was carried out, based on the Euler angles converted from 2D classification. “global” means global orientation alignment was carried out. “localnorm” means membrane normal vectors were used to constrain the orientation alignment (by setting “--sigma_tilt 10” in Relion4). For “local”, the initial reference was reconstructed directly using the Euler angles converted from 2D classification. For “global” and “localnorm”, only normal vectors were used to reconstruct the initial reference, the in-plane angles were set to random values.

-A more detailed comparison between MPicker and existing software, emphasizing its advantages would bolster the case for its adoption.

A: Thanks for the suggestion. The major differences between MPicker from other software are the membrane flattening and its functional flexibility in data processing. It is hard to make a detailed comparison among them because they are based on different principles. Nevertheless, we made **Table R1** to compare three software for particle picking on membranes. This table has been added

in the revised manuscript as **Extended Table 1**.

**Table R1 | Comparison of functions among three software.**

	MPicker	Membranorama	Dynamo catalogue
Inputs	Membrane mask, or labeled points, or triangle mesh	Triangle mesh	Labeled points
Generate flattened tomogram	✓	✗	✗
Generate triangle mesh	✓	✗	✓
Generate normal vector	✓	✓	✓
Label in-plane angle	✓	✓	✗
Render 3D surface with grey level	✓	✓	✗
Particle picking	In flattened tomogram	On 3D surface	In raw tomogram, or oversampling

-Briefly discussing potential future applications beyond the demonstrated examples, such as
integration with downstream analysis pipelines, could further solidify MPicker's significance for the
cryoET field.

350 A: Thanks for the suggestion. We added a paragraph in “**Discussion**” to further explore the issues
about membrane flattening, particle picking, and 2D classification. As follows:

*Currently, membrane extraction requires manual interaction and is the most time-consuming part*
*of MPicker. Automation is still needed, so that users can integrate the membrane flattening to their*
*workflows without much manual intervention. Moreover, improved particle picking methods that*
*can take full advantage of flattened tomograms have yet to be developed. In addition, improving the*
*2D classification of protein projections on the membrane surface may be useful, which should be*
*more computationally efficient than 3D classification in isolating small or unknown proteins from*
*a mess of mixed molecules on the membrane.*

Overall, this work by Yan et al. presents a promising software tool, MPicker, with the potential to
significantly improve cryoET workflows for studying organelles and membrane proteins.

Addressing the suggestions above would further strengthen the manuscript and solidify MPicker's
impact on the field.

365 A: Thanks for your suggestions. We have made corresponding revisions in the revised manuscript.

Reviewer #3 (Remarks to the Author):

The authors present a method for visualization and localization of membrane proteins in cryo-
electron tomograms. The core of the technique lies in membrane flattening which is very well
described including its pitfalls in the form of possible distortions. In general, the authors describe
all the essential steps in the workflow in a very clear manner and are critical of their shortcomings,
even providing quantification for their uncertainty in some cases. This is rarely seen in recent
method development/publications and the effort should be appreciated. The results show the
robustness of their method w.r.t. the different cells as well as different membrane morphologies and
orientations. The software provides a healthy mixture of automated and manual approaches for most
of the steps. Despite many advances in cryo-electron tomography, particle picking is still not
straightforward, and thus having tools that allow for easy manual corrections of automated
approaches is often crucial for successful data processing. I also appreciate the possibility of using
external inputs, e.g. membrane segmentation. MPicker has the potential to become a valuable tool
for membrane proteins processing. I recommend the manuscript for publication provided the authors
address the following points:

385 A: Thanks for your suggestions.

Major issue:

I could not find the STA maps produced within this study in the provided material. This is not
acceptable - the maps need to be provided at least to the reviewers and ideally uploaded to the
EMDB and released upon publication (in case they are connected to other unpublished manuscripts
they can be released later but their accession code should be provided here already). The sentence
about availability upon "reasonable request" cannot be meant seriously. On a similar note, the
achieved resolutions are not supported by FSC curves, authors should add them.

395 A: Apologies for missing the maps. We uploaded the STA maps together with the revised manuscript,
including the PBS and PSII-LHCII. Additionally, we have deposited the final PSII-LHCII map to
the EMDB with accession number EMD-61019 (will be released upon publication), and added the
information into the "Data availability" section of the revised manuscript.

Regarding the FSC curves, we added **Extended Data Fig. 10** (please see the copy below) to
display them and described their calculation in the "Calculation of subtomogram averaging"
section of the article.

**Extended Data Fig. 10 | FSC curves of 3D reconstruction of the PBS and PSII-LHCII**
**complex**

**Minor issues (in random order):**

1) The functionality provided by Membranorama seems very similar to MPicker. Authors should
provide more information on the differences between these two and possibly also the advantages of
MPicker over Membranorama (if any).

While there is some overlap in functionality between Membranorama and MPicker, visualizing
membrane surfaces and manually picking particles, the techniques and usage between them are
fundamentally different. We made **Table R1** to coarsely compare three software for membrane
protein particle picking. We also list some detailed differences between Membranorama and
MPicker below, based on our usage experience.

- 1) MPicker warps a sub-tomogram region based on the shape of the membrane surface,
generating a new flattened tomogram. Membranorama utilizes the raw tomogram to obtain
grey-level information on the membrane surface and renders it in 3D space. The key
advantage of MPicker is that there is more 3D information from the flattened tomogram than
from a simple 3D surface rendering.
- 2) MPicker uses a radial basis function (RBF) to represent the surface, and users only need to
provide a membrane mask or even manually annotated points. In contrast, Membranorama
requires users to provide a triangle mesh (such as an obj file) to represent the membrane
surface, which is not easy to obtain. MPicker also supports triangle mesh as the input.
- 3) By generating flattened tomograms, MPicker bypasses 3D rendering. Picking particles in
flattened tomograms is just like the particle picking in ordinary tools, accordingly, automated
picking becomes convenient. 3D rendering and triangulation also become simple on the basis
of the flattened tomogram, and are provided as an auxiliary function of MPicker.
Membranorama performs most operations (such as particle picking) in 3D, which makes the
operation in one more dimension, and hence not as straightforward and smooth as in 2D. And
only manual particle picking is supported in Membranorama.
- 4) MPicker can output a triangulated membrane surface (obj file), which carries the texture and
can be loaded by other software (such as the Windows 3D viewer or Blender, as well as
Membranorama).
- 5) MPicker generates a separate flattened tomogram for each membrane, allowing users to
process membranes one by one and easily visualize the relationship between adjacent
membranes. Moreover, flattened tomograms are ordinary MRC files, and users can use any
favorite software to pick particles or do other processing in the flattened tomograms. To our
knowledge, Membranorama doesn't well support processing individual membrane separately,
in the case of many membranes stacked together (such as **Fig. 4c**).
- 6) MPicker can also correlate and display the point positions on the fly between the raw and
flattened tomograms. By providing multiple views of the same region users can easily locate
and identify proteins. In contrast, Membranorama displays only the surface of the membrane,
corresponding to the xy slice of the flattened tomogram in MPicker.

7)

In summary, by splitting membranes into individual flattened tomograms, MPicker improved
the convenience and efficiency in membraned analysis, and provided high flexibility in data
processing. Furthermore, we provided a comparison among software, MPicker, Membranorama,
and Dynamo, and summarized their differences in the following **Table R1**. This table has been
added in the revised manuscript as **Extended Table 1**.

**Table R1 | Comparison of three tools that assist in particle picking of membrane proteins in**
**CryoET.**

	MPicker	Membranorama	Dynamo catalogue
Inputs	Membrane mask, or labeled points, or triangle mesh	Triangle mesh	Labeled points
Generate flattened tomogram	✓	✗	✗
Generate triangle mesh	✓	✗	✓
Generate normal vector	✓	✓	✓
Label in-plane angle	✓	✓	✗
Render 3D surface with grey level	✓	✓	✗
Particle picking	In flattened tomogram	On 3D surface	In raw tomogram, or oversampling

Adding a comparison on one tomogram might be interesting as well.

458 A: Although Membranorama may offer more options to optimize 3D rendering, the 3D display in
MPicker is sufficient in general, and the surface of MPicker can be exported (with texture). We
rendered the same surface with both tools, as shown in **Fig. R1**. Two software show nearly the same
display quality.

**Fig. R1 | Comparison of membrane drawing between MPicker and Membranorama. a,** The
 raw tomogram (the same as **Fig. 1a**) and an x - y slice of flattened tomogram (orange). The surface
 corresponding to the slice was exported as a triangle mesh and used as the input of Membranorama.
 **b,** The same surface in **a** is presented in 3D by MPicker and Membranorama, which illustrates the
 similar display quality.

2) Each membrane present in a tomogram is flattened separately (I assume) and the automatic
 picking using EPicker is performed on the slices of flattened tomograms.

473 A: Yes. The understanding is correct.

Given that, the 1616 false positives out of 5798 picks and 1144 missing particles is rather poor
 performance. Do the missing picks and false positives correlate to the regions of larger distortions?

478 A: The main reason causing this “unideal” result was that many unknown densities exist on the
 479 membrane, and we only chose the particles with possible features of PSII. That is, some of the
 480 removed particles might be good particles under the criterion of particle picking, which are removed
 just because these particles are not PSII. To better demonstrate the challenges, the following **Fig.**
 **R4** shows some non-optimal picking results, in which many unknown particles and complicated
 contrast are present. In addition, the statistics of the picking performance for PSII in the present
 work is: recall 0.79, precision 0.72, and F1 score 0.75, which is indeed not perfect but not a bad
 result compared with other published results.

Distortion is not the main issue because the distortions in this test are subtle. As shown in **Fig.**
 **R4** and **Extended Data Fig. 4**, no obvious distortion was found.

**Fig. R4 | Examples of suboptimal particle picking results from flattened tomograms.** Picking
 results of the current slice (z) and two nearby slices ($z \pm 1$) are plotted. The bottom part of the
 flattened tomogram in **b** is out of the region of the membrane, which can be seen in its side view
 (right).

Or is it rather due to the 2D slices having different properties than typical raw 2D micrographs?

499 A: Yes, the membrane slices have a complicated *in situ* environment, and hence contain many
unknown particles and image features, as well as poor SNR. This is a much more challenging
situation for particle picking than that of the purified proteins in typical raw 2D micrographs.

Was some other particle picking approach tried as well?

505 A: Thanks for the suggestion. We tested the Laplacian-of-Gaussian (LoG) method and 2D template
matching in RELION on the slice of the flattened tomogram, as well as 2DCNN-based particle
segmentation in EMAN2, as shown in **Fig. R5**. LoG did not work, despite various parameters were
tried. The results from template matching and 2DCNN look not bad.

**Fig. R5 | Particle picking results on flattened tomogram using other methods. a**, Picking results
using template matching (TM) and Laplacian-of-Gaussian (LoG) in RELION. The two selected
classes in **Fig. 5c** were used as TM references. **b**, The result of 2DCNN-based segmentation in
EMAN2, in the right panel the particles are highlighted by the segmentation. The left panel shows
the raw images.

Is there support to upload coordinates from other approaches that would correspond either to the 2D
slices or even 3D coordinates from raw tomograms (e.g. from 3D template matching on raw
tomograms)?

522 A: Yes, MPicker supports both of them. MPicker allows the import of coordinates through both GUI
and scripts. MPicker can convert coordinates between the flattened and raw tomograms.
Accordingly, users can pick particles either in flattened or raw tomograms using any other software,
and import the coordinates of picked particles to MPicker.

3) Does Relion2 mean Relion version 2? If so, why use so old version?

529 A: Yes, Relion2 is Relion version 2. The reason to use Relion2 is that its input/output data format is
530 simpler than the newer version, which simplifies the data conversion. We also tried to use Relion4
with various settings and found that the results were not improved and sometimes worse, as shown
in **Fig. R2**.

Nevertheless, we found that enabling the functions of “Tomo CTF refinement” and “Tomo
frame alignment” with fixed particle orientations in Relion4 improved the resulting map a little bit.
Consequently, we replaced the result in **Extended Data Fig. 6 a-b** with the new map and updated
the final resolution from 25.4 Å to 24.2 Å. We also updated the corresponding descriptions in the
sections "Orientation estimation of membrane proteins" and "Calculation of subtomogram
averaging" in the revised manuscript.

**Fig. R2 | STA of PSII-LHCII using different workflows. a**, The same density map as shown in

**Fig. 5e.** Local orientation alignment of Relion2 was applied on particles selected from 2D
classification, which generated the best result. **b,** The density map generated in the same workflow
as **a** but using Relion4. The structure is not as clear as that in **a**. **c-e,** Some results of 3D refinement
using “general” STA workflows. The structure features are unclear. **f,** Results of 3D classification
using Relion2. The structure features are unclear. Two of the five classes are empty. **g,** Results of
3D classification using Relion4. The structure features are unclear. One of the five classes is empty.
In all the panels, the notions are described as follows. Images with red boxes are x–y slices crossing
the extramembrane region. Images with green boxes are x–z slices crossing the volume center.
Images with blue boxes are y–z slices crossing the volume center. Images with black boxes are x–y
slices crossing the transmembrane region. “Relion2” or “Relion4” means the program used for 3D
refinement or 2D classification. “part” means only 1846 particles selected from 2D classification
(**Fig. 5c**) were used. “all” means all 5326 picked particles were used. “local” means local orientation
alignment was carried out, based on the Euler angles converted from 2D classification. “global”
means global orientation alignment was carried out. “localnorm” means membrane normal vectors
were used to constrain the orientation alignment (by setting “--sigma_tilt 10” in Relion4). For
“local”, the initial reference was reconstructed directly using the Euler angles converted from 2D
classification. For “global” and “localnorm”, only normal vectors were used to reconstruct the initial
reference, the in-plane angles were set to random values.

4) Starting on line 293: "In the raw tomogram, the direction of the missing wedge was identical for
all the particles (3D). For particle projections (2D) along the normal vectors of the membrane, the
corresponding missing wedge shown in each projection differed among particles." This is
misleading - the direction of the missing wedge is identical in cut-out subtomograms but after they
are oriented (i.e. in correspondence to the normals) and then the direction differs. I suggest to
rephrase this.

570 A: Thanks for the suggestion. We have rewritten several corresponding sentences in the revised
manuscript as the highlighted in follows:

*MPicker generates projections directly from raw tomograms to avoid the possible distortion and*
*interpolation errors induced during flattening. Furthermore, because the cryoET data suffer from*
*the missing wedge issue, the particle projection along the membrane normal vector may also be*
*affected by the missing wedge, so the missing wedge must be considered during the 2D classification*
*(Extended Data Fig. 5). We refer to RELION's 3DCTF model²⁰, which is used to process the*
*contrast transfer function (CTF) and missing wedge in the STA.*

5) It is not obvious in the main text whether the 2D classification is done on raw or flattened
tomograms - it becomes clear in methods, but I suggest stressing it also in the main text.

582 A: Thanks for the suggestion. We have clarified it in the revised manuscript as follows: *MPicker*
*generates projections directly from raw tomograms to avoid the possible distortion and*
*interpolation errors induced during flattening.*

6) On line 356, "making STA more robust" is not a proper terminology in this case - robust STA

would provide the structure without imposing a strong initial orientation. I would leave that out and
keep only the statement on efficiency.

590 A: Thanks for the suggestion. We deleted the “robust” in the revised manuscript.

7) In the methods section authors mention in a few places that the results were computed on 2x
downsampled/binning data. Why not bin even more? The membrane segmentation tends to work
better with more binning due to better contrast and less noise (the segmentations are smoother).
Would more binning reduce the need to simplify/downsample the point clouds/meshes later on?

597 A: Apologies for missing the binning parameters. The membrane segmentation, flattening, and
598 particle picking were performed on 3x binned tomograms (pixel size 10.89 Å). We added a
599 description of this in the "**Preparation of Tomograms**" section.

In this work, we binned tomograms to a pixel size of around 10 Å. More binning may lead to
too few pixels to accurately describe the membrane, and hence fail to distinguish two surfaces of
the same membrane for surface extraction. Moreover, too large pixel size will also cause too few
pixels to describe the protein features.

8) Would be helpful to add some time estimates for the individual automated procedures in relation
to the data size. This is mostly missing in the methods. Same for memory requirements and
computational power needed for MPicker in general.

609 A: Thanks for the suggestion. For EPicker, computation time was discussed in the "**Particle picking
by EPicker**" section. For membrane segmentation, we added a description of the computation time
in the "**Preprocessing of Tomograms**" section, as follows:

*For the tomogram in Fig. 2b (960 × 928 × 178 voxels), the segmentation took about 12 min.
The segmentation was run on a workstation with two RTX 2080Ti GPUs.*

For membrane flattening, we have added a new section titled "**Computational Efficiency**" to
discuss it. As follows:

*Membrane flattening is not a computationally intensive task, and hence MPicker requires just
minimal computer performance. All tasks using MPicker in this work can be run on a typical laptop,
except for the part of membrane segmentation and EPicker which require a general-purpose GPU.
The time required for membrane flattening is typically several seconds, depending on the area of
the membrane, the spacing of the interpolation points, and the surface projecting method (onto a
plane or a cylindrical surface).*

*As an example, to generate a flattened tomogram in Fig. 4d, MPicker took ~3 s on surface
extraction (5 seed points), ~8 s to generate a flattened tomogram (172 × 833 × 31 voxels, and
flattening through a cylindrical surface, and using 414 interpolation points with 12-pixels spacing).
For another smaller surface (144 × 393 × 31 voxels for the flattened tomogram), MPicker took ~2
626 s on surface extraction (2 seed points), and ~3 s on flattening through a cylindrical surface (using
218 interpolation points with 12-pixels spacing). All these tests were run with an Intel Xeon Gold
6132 CPU.*

MPicker doesn't involve much computing, typically, several seconds for each step (see **Table**

**R2** below). Memory requirements depend on the specific data. For example, an ordinary laptop with
 16 GB memory is sufficient to process the data used in the manuscript (a laptop with 8 GB should
 also be sufficient but was not tested). MPicker uses the “mmap” function provided by a mrcfile
 library to open mrc files (memory mapping), so there is no need to load the entire tomogram into
 memory, which reduces memory requirements. During flattening, memory is mainly used to store
 the flattened tomogram and the coordinate mapping table (which is three times the size of the
 flattened tomogram).

**Table R2 | Computation time for flattening with different settings on two surfaces.** In our
 work, the interpolation spacing around 12 pixels is small enough in most cases, and the surfaces
 were projected onto the cylinder or plane depending on the curvature of the surface. The thickness
 of the flattened tomogram (the third value of volume size) has little effect on calculation time. The
 test was performed on an Intel Xeon Gold 6132 CPU and was repeated three times for each
 setting.

	projection	interpolation spacing (point number)	volume size	time
surf1	cylinder	12 (414)	172 × 833 × 31	8.04 ± 0.45 s
	cylinder	12 (414)	172 × 833 × 61	9.62 ± 0.18 s
	cylinder	6 (1458)	173 × 834 × 31	14.52 ± 0.04 s
	plane	6 (1476)	178 × 817 × 31	11.84 ± 0.09 s
	plane	12 (420)	174 × 812 × 31	4.76 ± 0.03 s
surf2	cylinder	12 (218)	144 × 393 × 31	3.01 ± 0.03 s
	cylinder	12 (218)	144 × 393 × 61	3.56 ± 0.03 s
	cylinder	6 (782)	148 × 393 × 31	4.53 ± 0.04 s
	plane	6 (782)	148 × 391 × 31	3.23 ± 0.02 s
	plane	12 (219)	144 × 388 × 31	1.66 ± 0.04 s

9) Are NX and NZ swapped in the equation s5 (line 512)? Normals are computed as the cross
 product of tangent vectors to the surface, no?

650 A: Sorry for the mistake. Thanks so much. We corrected it in the revised manuscript. The following
 is the corrected equation:

$$\begin{cases} NX = \frac{\partial Y}{\partial u} \frac{\partial Z}{\partial v} - \frac{\partial Z}{\partial u} \frac{\partial Y}{\partial v} \\ NY = \frac{\partial Z}{\partial u} \frac{\partial X}{\partial v} - \frac{\partial X}{\partial u} \frac{\partial Z}{\partial v} \\ NZ = \frac{\partial X}{\partial u} \frac{\partial Y}{\partial v} - \frac{\partial Y}{\partial u} \frac{\partial X}{\partial v} \end{cases} \quad (s5)_{\ast}$$

10) In equation s8 on line 533, I assume that the authors mean the arc length up to y, in which case
 the bounds could be added from -infinity to y.

658 A: Thanks for the suggestion. We modified the equation to clarify the integrating range from y_{\min} to
659 y . y_{\min} is the bound of the tomogram and hence logically is -infinity.

$$\begin{cases} u(x, y) = u(x) = x \\ v(x, y) = v(y) = \int_{y_{\min}}^y \sqrt{1 + \left(\frac{d}{dy'} h(y')\right)^2} dy' \end{cases} \quad (s8)$$

11) For the tilt-series acquisition - how many tilt-series were collected and how many were used for
processing?

665 A: We have a dataset of Chlamydomonas with dozens of tomograms. We finally used four
tomograms for further processing, because they had a high number of thylakoid membrane pairs,
where PSII is primarily located.

Two small remarks on acquisition parameters - the defocus of -4 to -6 μm seems very high given the
thin lamellae; the bidirectional acquisition scheme is not optimal for STA.

672 A: The high contrast from high defocus is good to demonstrate MPicker. We also tested relatively
lower defocus. As an example, we flattened a thylakoid membrane in another lamella (unpublished)
with an average defocus of -3.0 μm , and an HIV virus-like particle (EMPIAR-10164) with a defocus
of -2.5 μm , the results are clear, as shown in **Fig. R6**. In addition, the data collection scheme doesn't
have a visible influence on the image contrast, hence shouldn't affect the MPicker, and both
tomograms in **Fig. R6** were collected using the dose symmetric scheme. Anyway, the contrast and
SNR of the raw tomogram are the essential factors that directly determine the performance of
MPicker.

**Fig. R6 | Flattening results of tomograms with relatively lower defocus than that used in the**
**manuscript. a,** Flattening result of a thylakoid membrane in corn. The defocus is -3.0 μm . **b,**
Flattening result of an HIV virus-like particle (same as **Extended Data Fig. 9b**). The defocus is -2.5
685 μm .

12) IsoNet was used for all preprocessing. To my knowledge this requires STA to be run to estimate
the missing wedge. This might create a bit of a chicken-egg problem. Authors should comment more
on how exactly the IsoNet was used.

691 A: Sorry for the confusion, this is not a chicken-egg problem. IsoNet does not require the STA and
692 was only used to enhance the tomogram for visualization purpose in MPicker. STA used the raw
tomogram, and was carried out after MPicker (particle picking and/or orientation determination in
MPicker).

For some membranes, the missing wedge is not a problem, in such cases some denoising methods
might be sufficient - was anything else than IsoNet tried? How does MPicker perform on non-
denoised tomograms, e.g. those produced with the fSIRT filter in IMOD?

700 A: In fact, IsoNet is not always needed by MPicker, because we found that in many cases MPicker
can display membranes with good quality directly on raw tomograms. To demonstrate this
experience, a comparison is shown in **Fig. R7**. For *Chlamydomonas*, we compared the flattening
results from raw tomograms reconstructed with WBP, fSIRT, and tools in the Warp package with
those further processed by IsoNet.

**Fig. R7 | Comparison of flattening results from raw tomograms with those further processed**
**by IsoNet. a, *Chlamydomonas* thylakoid sample as in Fig. 4c. b, *Chlamydomonas* ER sample as in**

**Extended Data Fig. 3c. c**, Red alga thylakoid sample as in **Fig. 4b. d**. *Cyanobacterium* thylakoid
sample as in **Fig. 2c**. “Raw” means the tomogram is unprocessed by IsoNet (just the map
downloaded from EMDB in **b-d**). “WBP” means the tomogram was reconstructed using WBP in
Etomo. “fSIRT” means the tomogram was reconstructed using SIRT-like filter (fake SIRT) in Etomo.
“Warp” means the tomogram was enhanced in Warp. “IsoNet” means the final tomogram processed
by IsoNet was used.

13) Would the method benefit from 3D-CTF corrections on tomograms, or at least 2D-CTF
correction on tilt-series?

719 A: The map was always binned as mentioned before, typically, to a resolution of about 20 Å (pixels
size of ~10 Å). At this resolution level, CTF correction has very little influence on the contrast of
the tomogram. Therefore, CTF correction of 3D tomogram or 2D tilt-series was usually not
necessary, and not used in the manuscript (we applied CTF deconvolution to some data just as a
preprocessing step of IsoNet).

14) On line 659 authors mention "image quality of tomograms". The quality of tomograms is ill-
defined - the quality depends on the purpose. In this case, authors mean high contrast and less noise
and they should define the term "image quality" as such before using it.

729 A: Thanks for the suggestion. We changed “image quality” to “contrast and SNR”.

15) I am not familiar with DeepTomo (line 668) - is there possibly a reference missing there?

733 A: DeepTomo is a new software developed by our lab for missing wedge correction and particle
picking in tomograms. We are preparing to submit the DeepTomo manuscript and will put it on the
pre-print website soon. MPicker uses part of the code framework from DeepTomo as an optional
tool for membrane segmentation.

16) As for the calculation of Euler angles, the second point in manual labeling might be difficult to
assign consistently even in the flattened tomograms and the issue can be even more complicated for
complexes with C-symmetry. In practice, when surfaces are used to generate the normals of the
particles, it is often enough to randomize the third missing (in-plane) angle and then to allow for
full in-plane rotation during STA (not cone though). This procedure provides a sanity check for the
quality of picking - if STA does not produce any reasonable average either the picking was incorrect
or the data quality is very low. It would be good to add this feature of randomization as an option at
this step.

747 A: Thanks for the suggestion. MPicker supports these functions. Users can specify whether to use
randomized in-plane orientation or measured in-plane orientation. MPicker gives users the
flexibility to work with coordinates, normal vectors, and in-plane angles.

Also - what Euler angles are used (what convention)?

753 A: MPicker uses the same Euler angle convention, ZYZ under global coordinate system, as Relion
(rlnAngleRot, rlnAngleTilt, rlnAnglePsi).

17) If I understood the webpage on MPicker correctly, the positions (and orientations) are outputted
in a custom text format. Is there a possibility to export them directly as a starfile for Relion?

759 A: Thanks for the suggestion. The custom text file contains all the necessary information to generate
a starfile for Relion. We added an option to output the results in Relion's star format in the latest
version of MPicker. The output star file contains: rlnCoordinateX, rlnCoordinateY, rlnCoordinateZ,
rlnAngleRot, rlnAngleTilt, rlnAnglePsi, and rlnClassNumber.

18) In Fig.3e: the lamina filaments are not visible and should not be pointed out. The same for the
polysomes - their visibility is in general random and is not connected to the orientation of the
membrane at all (and definitely not in the presented case). I suggest removing it from the legend as
well as from the main text since it is misleading.

769 A: The purpose of showing these details (filaments and polysomes) is to illustrate that we can see
the same things as those reported in the original literature (Fig. 3 of reference 13,
<https://doi.org/10.1126/science.aad8857>, a copy of this figure is shown below). All the descriptions
were based on the original literature, and we didn't add new information and analysis. We think the
description is necessary for the reader to understand what they saw. The point is that the generation
of high-quality images similar to those in the original literature is simpler in MPicker.

Figure Redacted

19) I suggest providing the tomogram in Fig.3d as a supplementary video.

778 A: Thanks for the suggestion. We added **Supplementary Video 1** in the revised manuscript, which
went through the flattened tomogram (**Fig.3e**) and simultaneously showed the corresponding
locations in the raw tomogram (**Fig.3d**).

20) In the extended data figure 1 - I suggest adding the corresponding raw tomogram next to b-d
instead of referring to them in the legend.

785 A: Thanks for the suggestion. We modified **Extended Data Fig. 1** as you suggested.

Reviewer #3 (Remarks on code availability):

I have only checked the website and found it well done for its purpose. There is an installation guide
and a tutorial one can follow. Currently, one has to download the software from the website - for the
reproducibility and versioning (as well as other reasons) the authors should upload it to github or
something similar. The installation should not be a problem since conda can be used and authors
provide the complete environment for download as well. However, I have not tested it.

797 A: Thanks for the suggestion. We used our website (thuem.net) as the official source of MPicker.
The code is now also available on GitHub through the address <https://github.com/thuem/MPicker>.

We thank all reviewers for their valuable comments and suggestions. Comments from reviewers are highlighted in grey background throughout the text.

Reviewer #1 (Remarks to the Author):

The revised MPicker manuscript has added several significant pieces of data, including the "unfolding" of tubular and spherical structures in the new Figure S9. These additions enhance the visualization of key findings, but in my view, the paper still faces major limitations.

First, there remains a lack of conceptual advance. The comparisons made to other software and the six arguments regarding implementation details represent only an incremental step forward. The authors have not addressed this issue in the revised manuscript, nor have they made any clear modifications to strengthen the description of their conceptual framework. If the authors believe there is a notable conceptual advancement, it would be highly beneficial to explicitly describe it in the text.

A: Thank you for your suggestion. We have expanded the **Introduction** section and the **Extended Table 1** (Comparison of functions among different software) in the revised manuscript to further solidify our conceptual advancements. In this revision, we have further highlighted the innovative concept of reducing the dimensionality of curved membranes from three-dimensional (3D) to two-dimensional (2D) space.

Although some sub-functions of MPicker may bear resemblances to those found in other existing software, a direct comparison of the full functionality is challenging, and the functionality of MPicker should be considered holistically. We updated the **Extended Table 1** to highlight the advancement of MPicker that provides a unique solution to the challenges associated with in situ membrane protein analysis.

Additionally, I find one of the responses to my previous comments misleading. The authors state "...The signal from the membrane is very strong and makes alignment unreliable. This might be the reason that there is still no deposited in situ structure of PSII-LHCII in the EMDB." While this statement is technically not wrong, it is misleading because a structure of PSII alone was reported as early as 2010 (<https://doi.org/10.1016/j.bbabbio.2010.11.007>), where particles clearly aligned. Although this study is old and the raw data is likely no longer available for benchmarking, it still needs to be cited.

A: Sorry for the confusion. Our intention was to demonstrate the challenges in calculating STA within cellular samples. We appreciate the suggestion, and have added one sentence to mention the previous work about PSII and cited them in the section **Orientation estimation of membrane proteins** in the revised manuscript.

A critical issue that remains is the lack of a quantitative evaluation of MPicker's performance for sub-volume averaging. This is one of the most interesting claims of the paper, yet there is no comparative data showing why MPicker outperforms conventional 3D methods for sub-volume averaging. The question is: "in which aspects is MPicker more beneficial for sub-volume averaging

than the conventional 3D methods?" The PSII structure at 24 Å resolution is noteworthy, but the resolution is moderate, and no baseline data is provided for comparison. Given that the authors claim "automated particle picking and coarse alignment of membrane proteins for sub-tomogram averaging," it is important to demonstrate an advantage in this area—whether it's in processing speed, the number of particles picked, better alignment precision, or other relevant metrics. Using datasets such as HIV GAG, influenza virion (already present in the paper), or retromer on membranes (EMPIAR-10631), ion channels on membrane vesicles could help illustrate these benefits.

A: Thanks for the suggestion. We have added more information to address the concerns in the revised manuscript. The explanations are as follows.

Firstly, we added some quantitative tests for the coarse alignment in a new **Extended Data Figure 11** of the revised manuscript. The results indicated that the initial 3D orientation measured by MPicker is essential to avoid misalignment and promote convergence in the global alignment. In the previous manuscript, we mentioned that the 3D global alignment by Relion failed for STA of PSII. Now, we did an investigation for all the possible settings of Relion, and got a roughly correct (still worse than that calculated based on MPicker) reconstruction when reducing the angular step of the global orientation search to 1.8°. After comparing the angle distributions respectively calculated by Relion2 and MPicker (Class2D followed by just local alignment), we find that the correctness of the in-plane rotation angle on the membrane surface is the key. Relion can always determine the direction of the normal vector well, but the estimation of the in-plane rotation angle strongly depends on the angular sampling step. We further compared the 3D reconstructions of particles from different subsets, based on the orientation parameters from Relion under different angular sampling steps and MPicker. MPicker's results are always the better. Briefly, the quantitative comparison demonstrated that MPicker's coarse alignment provides a robust estimation for the initial orientation of PSII, which helps to reduce trial-and-error for the settings in STA.

Secondly, for the automated particle picking, both EPicker integrated in MPicker and the flattened tomogram indeed make the particle picking on cellular membranes much easier and faster than other software. The published works provided too limited information and datasets in this aspect to do a detailed comparison. So it is hard to make a solid comparison based on the published information and add it into the revised manuscript. Here we just discuss our thinking about this comment. A work published in 2020 for thylakoid membranes (<https://doi.org/10.7554/eLife.53740>) just used 396 PSII particles picked by Membranorama software. This work didn't give a reason why so few particles were picked. Anyway, particle picking with MPicker is convenient and efficient. As shown in our manuscript, the number of picked particles is easy to extend to thousands or tens of thousands. Although we present just one example, the literature and practical experience abundantly demonstrate the significant challenges associated with particle picking and further automatic particle picking on membranes within tomograms. MPicker provides a significant progress in this aspect.

Thirdly, for automated picking densely-distributed particles on viruses, vesicles, and tubes, MPicker can do the picking but might not be the best choice. For such samples, the oversampling or symmetry-based picking has been well-established and may have better efficiency than MPicker. MPicker is optimized on general membranes in cells. We clarified this point in the **Discussion** section to avoid overstatement of the performance of MPicker.

Fourthly, the demonstration of MPicker using high-resolution 3D reconstruction is challenging. Resolving **membrane proteins** to a high resolution as high as near-atomic/atomic or sub-nanometer is still very challenging for *in situ* cryoET, which mostly depends on the data quality itself rather than the particle picking and the coarse alignment.

Minor note: The improved PSII structure should likely be moved into the main figure to provide better visibility within the manuscript.

A: Thanks for the suggestion. We modified the **Fig. 5** in the revised manuscript and added the PSII maps.

Reviewer #2 (Remarks to the Author):

The authors have addressed my comments and questions. Congratulations to all the authors on this great work.

A: Thanks for your time and efforts.

Reviewer #3 (Remarks to the Author):

The authors fully addressed all my questions and concerns, and all changes made to the manuscript further approved its readability. The authors showed the novelty of their approach and transparently compared it to the other existing software. From my perspective, if a method is novel, there is no need to outperform existing software to qualify for publication. Particle picking is an unresolved problem in cryo-electron tomography; any tool that can make this process easier, more accurate, or faster should be welcomed. I do not have any further concerns and can recommend the manuscript for publication.

A: Thanks for your time and efforts.

I have a few remarks to the authors. However, their addressing is not necessary for the acceptance of the manuscript:

The response on line 499: I was not referring to the different environments between the *in situ* cellular samples and micrographs of purified proteins. I meant more different signal properties of the data. The 2D micrographs are essentially projections of whole structures, i.e., they contain a complete footprint of the structure. A slice from a tomogram has only information from that slice (although potentially convoluted due to the missing wedge). This is a totally different type of information that is contained there. Therefore, I am not surprised that methods designed for particle picking on 2D micrographs might not perform as well for tomogram slices. This is also demonstrated by the good performance of the 2DCNN from EMAN2, which was trained exactly on the 2D slices, not projections. This may be something to consider for future development of

pipelines using MPicker.

A: Thanks for the suggestion. We agree with you. The slice from a tomogram is very different from the 2D projection of the whole protein plus its surrounding things in both signal properties and background environments. As mentioned in the discussion section, we are planning to train a better model to make the particle picking more suitable for the flattened tomograms.

Regarding the response on the CTF correction: It is true that the resolution of the results gets affected by the lack of CTF correction only at subnanometer ranges. However, the shape of the structures is affected by the delocalization of the signal regardless of the resolution (nicely demonstrated in Fig. 3 in <https://doi.org/10.1016/j.jsb.2017.07.007>). For that reason, it might also be worth considering for MPicker.

A: Thanks for the suggestion. We agree that the CTF correction can improve the tomogram quality in theory. For now, the reconstruction is not a part of the MPicker, but users can freely choose which tomogram to use. We will further investigate the influence of CTF correction on the visualization and particle picking.

The premise of MPicker is to provide user-friendly software, not just a method. Therefore, I would strongly recommend adding direct support for more coordinate formats. I found the argument that Relion2 was used because it has easier inputs and outputs funny. The authors were able to implement MPicker with advanced GUI; I am sure adding a few parsers for loading star files and corresponding writer functionality should not be problematic and would improve user experience. Installing one more software just to perform the conversion is annoying, and the chances of someone using Relion2 are very low.

A: Thanks for the suggestion. We agree that conversions between software are always annoying for users. We are collecting the details of more coordinate formats, and will update the software soon.

Finally, in situ structures of phycobilisome and PSII are resolved at high resolution: <https://doi.org/10.1038/s41467-023-36175-y>.

A: This work got the high resolution by single particle method rather than STA. Achieving high-resolution by STA is still challenging for most membrane proteins.

Reviewer #3 (Remarks on code availability):

There were no changes in the code, so I stand by my previous statement.

A: We had updated the software according to the reviewer's comments and suggestions, and uploaded MPicker 1.1.0 to Github: <https://github.com/thuem/MPicker.git> and our website <https://thuem.net>.

We thank all reviewers for their valuable comments and suggestions. Comments from reviewers are highlighted in grey background throughout the text.

Reviewer #1 (Remarks to the Author):

The revised version of the M-picker manuscript went through minor improvements. It is a good technical manuscript and the software certainly can be useful for further investigations by the cryo-ET community.

Nonetheless, my major concern from the previous round of revisions has not been addressed

“A critical issue that remains is the lack of a quantitative evaluation of MPicker's performance for sub-volume averaging. This is one of the most interesting claims of the paper, yet there is no comparative data showing why MPicker outperforms conventional 3D methods for sub-volume averaging. The question is: "in which aspects is MPicker more beneficial for sub-volume averaging than the conventional 3D methods?" The PSII structure at 24 Å resolution is noteworthy, but the resolution is moderate, and no baseline data is provided for comparison. Given that the authors claim "automated particle picking and coarse alignment of membrane proteins for sub-tomogram averaging," it is important to demonstrate an advantage in this area—whether it's in processing speed, the number of particles picked, better alignment precision, or other relevant metrics. Using datasets such as HIV GAG, influenza virion (already present in the paper), or retromer on membranes (EMPIAR-10631), ion channels on membrane vesicles could help illustrate these benefits”

A: Thank you for your comments. To reach atomic or near-atomic resolution by STA should depend on the data quality rather than MPicker which is just involved in the beginning step of a STA workflow. Finding suitable datasets and performing high-resolution tests is not a simple task, and still very challenging for the field. Nevertheless, we believe that this does not affect the conclusions of the present work. As mentioned earlier, we provided quantitative tests for MPicker's Class2D-based coarse alignment (the section **Subtomogram averaging of PSII without initial orientations** and **Supplementary Figure 8**), showing that it significantly improved the initial 3D orientation, which helped avoid misalignment in subsequent alignment steps. We also highlighted that both the flattened tomogram and the EPicker integrated in MPicker could facilitate faster and more efficient particle picking on membranes.

If the authors are to publish the paper, they should deposit the tomograms and the positions of PSII particles that were used for an unsuccessful subtomogram alignment attempt with Relion2 (lines 344 onwards). EMPIAR would be a good platform for it.

A: Thank you for your suggestion. We deposited the raw data of PSII to the EMPIAR, as well as the coordinates and initial orientations before and after the 2D Classification.

Minor:

I would not call the extraction of a membrane to 2D to be a fundamental concept, as said in line 84

of the updated MS.

Line: 1244 : “This relies on the software THUNDER for the 2D classification”. Each software will have the classification functionality if external software is used, probably all boxes should be ticked.

A: Thank you for your suggestion. In the revised manuscript, we deleted the row “classification” in **Supplementary Table 1** to avoid possible misunderstanding. However, we need to clarify that the 2D classification played an important role in our work and could not be replaced by conventional 2D or 3D classification. As far as we know, currently, no other software can provide 2D projections in tomograms with the specified orientation and thickness. Moreover, a special feature of MPicker is the conversion from the result of 2D classification to the orientation and shift in 3D.

Reviewer #2 (Remarks to the Author):

The authors have addressed my comments and questions and I think this manuscript should be published. Congratulations to all the authors.

A: Thanks for your time and efforts.

Reviewer #3 (Remarks to the Author):

As already stated before, from my perspective, the authors provided satisfactory answers, and the manuscript is ready to be published.

A: Thanks for your time and efforts.

A final note to the authors: the difference between in situ and in vitro structure does not come from the averaging technique used but from the sample. Thus, the structure obtained by GISPA counts as an in situ structure of the phycobilisome because it was obtained from a crowded cellular environment, not from a purified sample.

A: Apologies for the misunderstanding. We agree that the GISPA is a good method to obtain in situ structures. We just wanted to say that MPicker was designed only for tomogram data and could not deal with GISPA data for now. From a biological view, there are still many differences between the red algae PSII-phycobilisomes and the Chlamydomonas PSII-LHC.

Reviewer #3 (Remarks on code availability):

Commented on before.